# TRION: FFT-BASED DYNAMIC SUBSPACE SELECTION FOR LOW-RANK ADAPTIVE OPTIMIZATION OF LLMS

**Ionut-Vlad Modoranu**[*]
ISTA[†] & Together AI

**Mher Safaryan**
ISTA[‡]

**Erik Schultheis**
ISTA

**Max Ryabinin**
Together AI

**Artem Chumachenko**
Together AI

**Dan Alistarh**
ISTA & Red Hat AI

## ABSTRACT

Low-rank optimization has emerged as a promising direction in training large language models (LLMs) to improve running time and reduce the memory usage of adaptive optimizers by constraining learning to a lower-dimensional space. Prior work typically projects gradients of linear layers using approaches based on Singular Value Decomposition (SVD) or QR-decomposition. Applying these techniques individually to each layer in large models is computationally expensive and incurs additional memory costs due to storing the projection matrices. In this work, we propose a computationally efficient and conceptually simple, two-step procedure to approximate SVD/QR-based gradient projections into lower-dimensional spaces by using a predefined orthogonal matrix of the Discrete Cosine Transform (DCT). We dynamically select columns from the DCT matrix based on their alignment with the gradient of each layer. The effective projection matrices are obtained via a simple `matmul` with the DCT matrix in $O(n^3)$ time, followed by a lightweight sorting step to identify the most relevant basis vectors. For large layers, DCT can be computed via `Makhoul`'s $N$-point algorithm based on Fast Fourier Transform (FFT) in $O(n^2 \log(n))$ time, yielding speed-ups for low-end GPUs. Due to the predefined nature of the orthogonal bases, they are computed once at the start of training. Our numerical experiments on both pre-training and fine-tuning tasks demonstrate the effectiveness of our dual strategy in approximating optimal low-rank projections, obtaining an approach with rank-independent running time that matches the performance of costly SVD/QR-based methods while achieving faster runtime and reduced memory usage by up to $25\%$ across different model sizes. Our code is available at https://github.com/IST-DASLab/Trion.

## 1 INTRODUCTION

The Adam optimizer (Kingma & Ba, 2014) and its regularized version AdamW (Loshchilov & Hutter, 2019) have become the standard approach for optimizing deep neural networks in various settings. With the recent increase in scale of LLMs up to billions and trillions of parameters, training with AdamW becomes more and more challenging, as its internal state requires two momentum buffers that scale with the model size. This practical problem paved the way for a line of research that focuses on reducing the memory usage of optimizer states in the context of adaptive gradient optimization. These approaches range from quantizing the states to 8 bits (Dettmers et al., 2021) to the recent GaLore optimizer (Zhao et al., 2024) inspired from the LoRA techniques (Hu et al., 2021; Lialin et al., 2023), that compresses the gradient matrix using low-rank decomposition based on SVD. Several improvements to GaLore have been proposed to enhance its performance, such as LDAdam (Robert et al., 2025), FRUGAL (Zmushko et al., 2024), FIRA (Chen et al., 2024), BAdam (Luo et al., 2024), Q-GaLore (Zhang et al., 2024). The key aspect of GaLore and its later improvements is the low-rank decomposition based on matrix factorization, such as SVD or QR decomposition.

---

[*]Correspondence to `ionut-vlad.modoranu@ista.ac.at`. Part of work done during an internship at Together AI.

[†]Institute of Science and Technology Austria

[‡]Current affiliation: Lancaster University, UK

Recently, the Muon (Jordan et al., 2024) optimizer sparked the community's attention due to the faster convergence in pretraining settings, achieved by orthogonalizing the momentum matrix using Newton-Schulz, an iterative procedure difficult to parallelize for large-scale pretraining runs *as the full-sized matrices must be materialized* on GPU. The Dion optimizer (Ahn et al., 2025) aims to reduce this overhead by employing low-rank, orthogonal updates. However, it requires storing a projection matrix for each layer and uses QR-decomposition to orthogonalize the low-rank components, making its running time dependent on the rank.

All these SVD/QR-based techniques are known to be computationally intensive as they have to be invoked for each linear layer, either at each step (for, e.g., LDAdam, Muon) or once at a few steps (for, e.g., GaLore). To mitigate the high computational and memory costs of these procedures, we ask: *can we find an alternative low-rank projection approach to serve as an accurate replacement for the orthogonal matrices in SVD/QR for low-rank compression of optimizer states, that is much cheaper to compute, and portable across memory-efficient optimizers?*

**Contributions.** In this work, we address our research question by proposing a cheaper alternative to the orthogonal matrices computed via SVD/QR. Concretely, we propose using the orthogonal matrices from the general class of Discrete Fourier Transforms, such as the Discrete Cosine Transform (DCT), which has been successfully used in image compression for the JPEG algorithm. To the best of our knowledge, we are the first to use it in the context of low-rank adaptive gradient optimization. We summarize our contributions as follows:

- We propose a dynamic column selection approach to adaptively choose columns from a fixed orthogonal matrix to compute a low-rank projection of the input matrix. The effective projection matrix is obtained from the fixed orthogonal matrix by indexing the columns (Section 2.1);

- We motivate that DCT is a good candidate for our dynamic column selection approach due to the reduced time complexity to compute the alignments with the input matrix via the Makhoul's $N$-point algorithm (Makhoul, 1980) to compute a fast DCT with $O(n^2 \log(n))$ time complexity compared to $O(n^3)$ incurred by a basic matrix multiplication. This can yield speedups of up to $8 - 50\times$ in computing the alignments for large layers (Appendix C) on low-end GPUs.

- We theoretically justify our dynamic column selection approach to compute the most significant columns of any fixed orthogonal matrix (including the DCT-based) to obtain a projection matrix tailored to each layer (Section 4);

- We show the DCT-based projection is a fast and accurate replacement for the orthogonal matrices computed via SVD/QR for the Muon- and Adam-like optimizers (such as Dion, FRUGAL, FIRA, LDAdamW) in the context of low-rank compression of optimizer states. We reduce the running time and improve the memory usage as we store only one DCT matrix per GPU for the entire network, computed once at the beginning of the training. In addition, we store only $r$ (rank) integers, representing the indices of the most significant columns for each layer instead of storing one low-rank projection matrix.

- We propose two standalone optimizers that use the DCT-based dynamic column selection approach:

  1. **Trion** (Section 2.4), which improves Dion by replacing the Power-Iteration with our DCT-based dynamic column selection approach to compute a low-rank representation of the momentum buffer, followed by orthogonalization via Newton-Schulz iteration. We provide a DDP-compatible implementation that communicates orthogonal, low-rank momentum across devices and compute the final layer updates locally using the projection matrix obtained from DCT. To the best of our knowledge, we are the first ones to reduce the complexity of Newton-Schulz using a low-rank approximation of momentum.

  2. **DCT-AdamW** (Section 2.5), which replaces the SVD-based low-rank projections and optionally adds quantized error feedback for the projection error. Further, DCT-AdamW rotates the momentum buffers such that new low-rank gradients are correctly incorporated, and thus allows changing the low-rank subspace every step.

## 2 METHOD

This section presents our dynamic column selection approach that works with any orthogonal matrix, briefly introduces the Discrete Cosine Transform (DCT) and the motivation of using it and finally

introduces the two algorithms we propose: Trion as an improvement to Dion to replace Power-Iteration and DCT-AdamW as an improvement to low-rank AdamW variants to replace SVD/QR.

## 2.1 Dynamic Column Selection

**General View.** Given an orthogonal matrix $Q \in \mathbb{R}^{n \times n}$, we want to compute a projection matrix $Q_r \in \mathbb{R}^{n \times r}$ by selecting $r$ columns from $Q$ to project the gradient $G \in \mathbb{R}^{n \times n}$ to an $r$-dimensional space $g = GQ_r \in \mathbb{R}^{n \times r}$. First, we compute the similarities matrix $S = GQ$ containing the scalar products between rows of $G$ and columns of $Q$. We rank the columns of $S$ based on their $\ell_1$- or $\ell_2$-norm and then pick the indices of the largest $r$ columns according to the chosen norm, which we use to index columns in $Q$ to obtain $Q_r$. The $i^{th}$ column in matrix $S$ contains the scalar products between each row of $G$ and $i^{th}$ column of $Q$, as detailed in Appendix B. We view these scalar products as similarities (or alignments) of rows in $G$ with columns of $Q$ and we want to choose the columns of $Q$ with largest similarity with rows of $G$. Using this approach, we ensure a dynamic mechanism to choose the most appropriate columns of $Q$ for the current gradient matrix $G$ to minimize the projection error (see Section 4). Each layer will have its own set of $r$ indices for the columns. The dynamic behavior comes from the $n$-choose-$r$ possible sets of indices to select from.

The rule of thumb in the low-rank factorization methods for optimization we are targeting in this work is to compress the smallest dimension of a matrix to $r$ dimensions (e.g. from $\mathbb{R}^{n \times m}$ to $\mathbb{R}^{n \times r}$ for $n \geq m$). Usually, the smallest dimension of the gradient matrix is $d_{\text{model}}$, the hidden (embedding) size of the model. As a result, the memory overhead of our dynamic column selection approach is the cost of storing only one orthogonal matrix $Q \in \mathbb{R}^{d_{\text{model}} \times d_{\text{model}}}$ (DCT in our case) per GPU for the entire model and $r$ integers for the corresponding column indices from $Q$ for each layer to create $Q_r$.

## 2.2 Projection Types

We follow the standard projection approach introduced in GaLore, where for a gradient $G \in \mathbb{R}^{R \times C}$ ($R$ rows and $C$ columns) we choose to collapse the smallest dimension to rank $r$. This set of rules applies to each layer by choosing the left or right projection based on the layer dimensions.

For $R \geq C$, we apply a **right projection** and we choose the most appropriate $r$ columns from the DCT matrix of shape $(C, C)$, resulting a projection matrix $P \in \mathbb{R}^{C \times r}$ and a low-rank gradient $g \in \mathbb{R}^{R \times r}$. For $R < C$, we apply a *left projection* with DCT matrix of shape $(R, R)$, resulting a projection matrix $P \in \mathbb{R}^{r \times R}$ and a low-rank gradient $g \in \mathbb{R}^{r \times C}$.

## 2.3 Discrete Cosine Transform

The Discrete Cosine Transform (DCT) is widely used in the signal processing and data compression literature (e.g., JPEG algorithm for image compression) and consists of $n$ orthogonal basis vectors whose components are cosines (Strang, 1999). There are minor variations of DCT and in this work we will use DCT-II/III. We denote the DCT-III matrix of order $n$ by $Q \in \mathbb{R}^{n \times n}$, defined as $Q_{ij} = \sqrt{2/n} \cdot \cos \frac{i(2j+1)\pi}{2n}$, with $i, j \in [n]$, where the first row has to be divided by $\sqrt{2}$ in order for $Q$ to be orthogonal, i.e. $Q^\top Q = I_n$. The DCT-II matrix is the transpose of DCT-III. In Appendix A, we provide details on how we can efficiently materialize the DCT matrix on GPU. Once we compute $Q$, we can use it in the dynamic column selection approach to efficiently compute low-rank projections of any two-dimensional matrix.

In Appendix C, we discuss why we choose DCT matrix. In short, the particular structure of DCT allows us to enable fast computation for the similarities matrix $S$ in $O(n^2 \log(n))$ time using the Makhoul's $N$-point algorithm (see Appendix D) instead of $O(n^3)$ time for basic matmul.

## 2.4 Trion: DCT-based improvement to Dion

In this section, we present how we can apply the DCT-based dynamic column selection approach to compute low-rank projection of momentum in the context of Dion (Ahn et al., 2025) to obtain a faster and more accurate optimizer, which we call Trion, presented in Algorithm 1.

Dion uses Power-Iteration to compute a low-rank projection and orthogonalizes it using QR-decomposition, whose running time depends on the rank $r$. Instead, we propose replacing these techniques with a rank independent approach to compute the low-rank projection based on DCT matrix, which requires computing the similarity matrix $S_t$, followed by a top-$r$ ranking to determine the indices of columns that best align with the momentum matrix, denoted by the set $i_t$. The matrix $S_t$

represents the DCT of the momentum matrix and it can be computed using the Makhoul's algorithm or simply by only one matrix multiplication $S_t = B_t D_C$, where $B_t \in \mathbb{R}^{R \times C}$ is the momentum and $D_C \in \mathbb{R}^{C \times C}$ is the DCT matrix.

Once we determine the indices $i_t$ of the most significant columns in $D_C$, we can extract the low-rank momentum $b_t$ from the similarity matrix $S_t$, as well as the projection matrix $Q_t$ by indexing $D_C$, which will be used in (a) computing the projection error $\Delta_t$ and (b) projecting the orthogonal low-rank momentum back to the higher dimensional space to update the model.

We would like to emphasize that we input the low-rank momentum $b_t \in \mathbb{R}^{R \times r}$ to Newton-Schulz and not the original momentum buffer $B_t \in \mathbb{R}^{R \times C}$, which significantly reduces the computational overhead. Moreover, we can use the efficient triton kernels provided in the Muon implementation from the official Dion repository[1] to speed up the computations even further for large ranks $r$, as Newton-Schulz will operate with $r \times r$ matrices.

**Communication in Distributed Training.** We develop the Trion optimizer on top of the published code for the Muon [2] optimizer that leverages the ZeRO (Rajbhandari et al., 2020) approach in Distributed Data Parallel (DDP) settings. Specifically, the model update $O_t$ for a layer is computed only on one GPU, and the result is communicated to other GPUs using `all-gather`, drastically reducing the computation costs for large models (under the assumption that communication is cheaper than computation itself). Since we replicate the DCT matrix on each GPU, we would like to emphasize that we communicate only low-rank terms $o_t$ from the source GPU (the one computing the update) to other devices and perform the step $O_t = o_t Q_t^\top$ locally on each device. This way, we do not communicate full size, orthogonal matrices $O_t$, but only low-rank versions $o_t$, thus reducing the overall iteration time.

This is particularly useful in the FSDP-compatible implementation, where we can materialize the full low-rank matrices $o_t$ on each device using `all-to-all` (see this Muon implementation [3]) to perform Newton-Schulz on a low-rank input, followed by a resharding. After resharding, each GPU can compute the individual slice of $O_t$ and update its own shard. However, the FSDP implementation requires deciding how each layer should be sharded based on whether that particular layer requires a left- or right-projection in order to avoid unwanted calls to communication primitives to materialize full tensors to compute the alignments $S$.

---

**Algorithm 1** Trion Optimizer

---

1: Define the DCT-II/III matrix $D_C \in \mathbb{R}^{C \times C}$
2: **for** $t = \{1, 2, \ldots, T\}$ **do**
3:      $G_t = \nabla_\theta L(\theta_t) \in \mathbb{R}^{R \times C}$
4:      $B_t = M_{t-1} + G_t \in \mathbb{R}^{R \times C}$
5:      $S_t = \text{MAKHOUL}(B_t) \in \mathbb{R}^{R \times C}$              ▷ or $S_t = B_t \cdot D_C$ to compute similarities;
6:      $i_t = \text{DYNAMICCOLUMNSELECTION}(S, r) \in \mathbb{N}^r$ ▷ select largest $r$ columns by $\ell_1/\ell_2$-norm
7:      $Q_t = D_C[:, i_t] \in \mathbb{R}^{C \times r}$        ▷ projection matrix containing most significant $r$ columns
8:      $b_t = S_t[:, i_t] \in \mathbb{R}^{R \times r}$        ▷ extract low-rank momentum from the similarity matrix $S$
9:      $\Delta_t = B_t - b_t Q_t^\top$        ▷ the error is $b_t Q_t^\top$ and replaces $P_t R_t^\top$ error from Dion
10:     $M_t = \mu B_t + (1 - \mu)\Delta_t = B_t - (1 - \mu)b_t Q_t^\top$    ▷ update momentum with error feedback
11:     $o_t = \text{NEWTONSCHULZ}(b_t) \in \mathbb{R}^{R \times r}$       ▷ Newton-Schulz applied on low-rank $b_t$
12:     $O_t = o_t Q_t^\top \in \mathbb{R}^{R \times C}$     ▷ project orthogonalized low-rank momentum to original size
13:     $\theta_{t+1} = (1 - \lambda \eta_t)\theta_t - \eta_t \max(1, \sqrt{R/C}) O_t \in \mathbb{R}^{R \times C}$
14: **end for**

---

## 2.5 DCT-ADAMW

We propose DCT-AdamW, a standalone low-rank version of AdamW with DCT-based projection that has the option to use quantized error feedback (EF) (Seide et al., 2014; Karimireddy et al., 2019; Alistarh et al., 2018) and ensures momentum buffers integrate gradients from the same lower dimensional subspaces in a similar way as in LDAdamW (Robert et al., 2025). In contrast to LDAdamW, which has to store two consecutive projection matrices per layer, we only have to store

---

[1] github.com/microsoft/dion
[2] https://github.com/KellerJordan/Muon/blob/master/muon.py#L138
[3] github.com/microsoft/dion/blob/main/dion/muon.py#L22

two sets of $r$ indices per layer. Prior work MicroAdam (Modoranu et al., 2024) quantized the error feedback down to 4-bits in the context of compressing the optimizer state using sparsity. In our setup, the lowest resolution we can quantize EF to is 8-bits without degrading the optimizer performance. For space constraints reasons, we present the pseudocode of our DCT-AdamW optimizer in Appendix G.

## 3 Experiments

In this section we present our numerical results. Our main goal is to show that our DCT-based dynamic column selection approach at least recovers the performance of the original algorithms which we integrate it in. Specifically, we evaluate Trion against Dion, where we directly compare the DCT-based projection followed by Newton-Schulz iteration with QR-based Power-Iteration. In addition, we replace the SVD-based projection in FRUGAL and FIRA optimizers with our DCT approach. In the end, we compare LDAdamW with our standalone DCT-AdamW optimizer. In our pretraining experiments, we compare training/validation perplexity and for fine-tuning we compare the evaluation accuracy, as well as memory usage and running time for both scenarios. In the remaining part of the paper, we focus on presenting our pretraining (PT) results and the fine-tuning (FT) results are presented in Appendix K.

We train from scratch models from the Llama family with 350M, 800M and 1.3B parameters using Chinchilla-optimal token counts (20 tokens per parameter) from the C4 (Raffel et al., 2020) dataset and sequence length 512. All PT experiments are run in Distributed Data Parallel (DDP) settings on 8x H100 Nvidia GPUs using global batch size 512 with local batch size 64 per GPU (unless otherwise specified explicitly).

**PT with Trion.** Our purpose is to show that our DCT-based dynamic column selection approach can successfully replace the QR-based Power-Iteration procedure in Dion and at least recovers the performance for the best hyper-parameters reported by the original work. In Table 1, we present our results, which are obtained using the optimal learning rate $\eta = 0.01$ (as reported by the original Dion paper) and weight decay $\lambda = 0.01$. We report the average values across 3 seeds for train/validation loss and perplexity, maximum allocated memory (read directly from the GPU) and the lowest running time across all runs. We experiment with ranks $128, 256$ and $512$, the rank-to-dimension ratio is $r/d \in \{1/16, 1/8, 1/4, 1/2\}$, equivalent to $6.25\%, 12.5\%, 25\%$ and $50\%$ of the full rank $d$, where $d$ is model's embedding dimension.

**Performance** In Table 1, we show Trion consistently achieves lower training and validation loss, which also translates to lower perplexities. At the top row of Figure 1 we show the training loss curve of Trion is lower than Dion across iterations. This is an indication that our DCT-based dynamic column selection algorithm, followed by orthogonalization via Newton-Schulz can successfully replace the Power-Iteration procedure in Dion.

**Memory Usage.** In Table 1, we show Trion has a lower memory requirement across all experiments since it allocates only one DCT matrix per GPU of size $d \times d$, from which we select $r$ columns using the indices of the most significant columns. In contrast, Dion stores a projection matrix for each layer. This design difference translates to around $10\%$ lower memory footprint for Trion compared to Dion.

**Runtime.** We would like to emphasize that the running time of Trion does not depend on rank, as can be seen from the reported runtime across different ranks in Table 1: Trion achieves nearly constant runtime for each model size across all ranks, while the runtime of Dion clearly depends on the rank because of the QR-decomposition. This represents an advantage of Trion for much larger scales compared to Dion. At the bottom row of Figure 1 we present the wall clock time for the two optimizers for rank 256 across all three models we tested on. Concretely, for a fixed running time budget, Trion consistently achieves lower training loss than Dion. Trion is faster than Dion by $2.5 - 4.5\%$ for rank $128$, $4.5 - 9\%$ for rank $256$ and $8 - 18\%$ for rank $512$ (the overhead of Dion increases with rank). It is important to mention that the embedding size $d$ of the models we used in our work is too small to see a significant difference in the runtime between Makhoul's algorithm and matmul when computing the column similarities. However, our benchmark presented in Appendix D shows the advantage of using Makhoul's algorithm at scale compared to matmul.

**Additional Runtime Benchmarks.** In Appendix F we provide a more comprehensive ablation for the optimizer step by comparing full-rank Muon with Dion and Trion to show the behavior of our Dynamic Column Selection algorithm when computed via `matmul` and via `Makhoul`'s algorithm. It is important to mention that the benefits of Makhoul's algorithm can be seen on older-generation

| Rank $r$ | Metric | 350M ($d=1024$) | | 800M ($d=2048$) | | 1.3B* ($d=2048$) | |
|---|---|---|---|---|---|---|---|
| | | Trion | Dion | Trion | Dion | Trion | Dion |
| | $r/d$ | 1/8 | | 1/16 | | 1/16 | |
| | Train Loss | **2.726** | 2.764 | **2.532** | 2.555 | **2.475** | 2.503 |
| | Train PPL | **15.29** | 15.89 | **12.59** | 12.89 | **11.90** | 12.24 |
| 128 | Val Loss | **2.736** | 2.773 | **2.505** | 2.527 | **2.422** | 2.448 |
| | Val PPL | **15.43** | 16.00 | **12.25** | 12.52 | **11.27** | 11.56 |
| | Memory (GB) | **42.42** | 45.56 | **67.45** | 71.64 | **63.62** | 68.57 |
| | Runtime | **1h 53m** | 1h 58m | **8h 5m** | 8h 24m | **19h 13m** | 19h 42m |
| | $r/d$ | 1/4 | | 1/8 | | 1/8 | |
| | Train Loss | **2.715** | 2.741 | **2.533** | 2.546 | **2.476** | 2.492 |
| | Train PPL | **15.11** | 15.51 | **12.61** | 12.78 | **11.92** | 12.11 |
| 256 | Val Loss | **2.727** | 2.750 | **2.503** | 2.519 | **2.423** | 2.440 |
| | Val PPL | **15.30** | 15.64 | **12.22** | 12.42 | **11.28** | 11.47 |
| | Memory (GB) | **42.42** | 45.59 | **67.45** | 71.75 | **63.62** | 68.58 |
| | Runtime | **1h 53m** | 2h 3m | **8h 3m** | 8h 36m | **19h 13m** | 20h 6m |
| | $r/d$ | 1/2 | | 1/4 | | 1/4 | |
| | Train Loss | **2.708** | 2.718 | **2.528** | 2.535 | **2.470** | 2.481 |
| | Train PPL | **15.01** | 15.16 | **12.54** | 12.63 | **11.84** | 11.97 |
| 512 | Val Loss | **2.718** | 2.727 | **2.499** | 2.509 | **2.420** | 2.430 |
| | Val PPL | **15.15** | 15.29 | **12.18** | 12.30 | **11.25** | 11.36 |
| | Memory (GB) | **42.42** | 45.99 | **67.45** | 71.81 | **63.62** | 68.73 |
| | Runtime | **1h 54m** | 2h 14m | **8h 3m** | 8h 48m | **19h 13m** | 20h 44m |
| | Train Loss | 2.697 | | 2.517 | | 2.513 | |
| | Train PPL | 14.85 | | 12.41 | | 12.35 | |
| | Val Loss | 2.707 | | 2.489 | | 2.409 | |
| Muon | Val PPL | **14.99** | | **12.05** | | **11.13** | |
| | Memory (GB) | 42.42 GB | | 67.45 GB | | 63.64 GB | |
| | Runtime | 1h 52m | | 8h 25m | | 19h 17m | |

Table 1: Perplexity, Memory, and Running Time Comparison for Trion and Dion. Muon is added for reference. $d$ stands for the embedding dimensionality of the model. The 1.3B model was trained with local batch size 32.

GPUs, which have fewer/less powerful tensor-cores. However, on newer GPUs, the tensor-cores are much faster than the gains of FFT-based procedures. We benchmarked `matmul` and `Makhoul`'s algorithm on Nvidia H100 and RTX 2080/3090 and provide details in Appendix E.

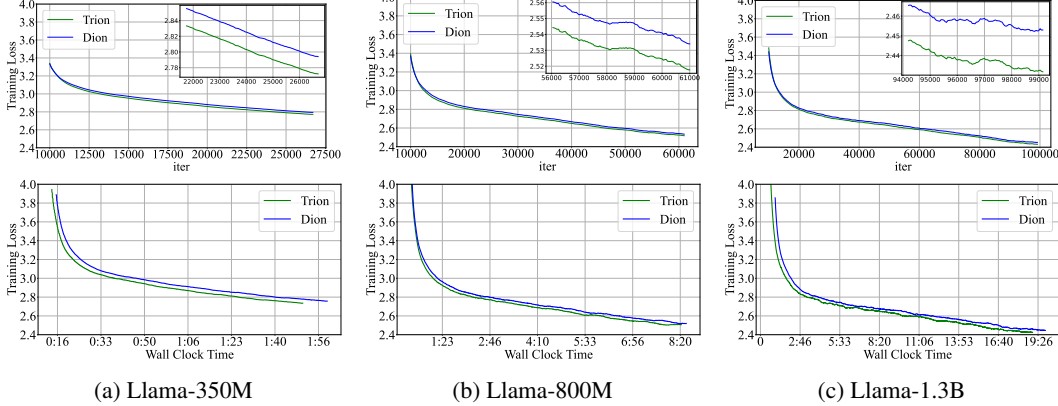

| (a) Llama-350M | (b) Llama-800M | (c) Llama-1.3B |
|---|---|---|

Figure 1: Training Loss across iterations (top row) and wall clock time (bottom row) for three Llama models trained with Dion and Trion optimizers on C4 using 20 tokens per parameter and rank $r = 256$. For a clearer visualization, the training loss curve was smoothed using a moving average with a window size of 200 and we zoom-in on the last 5000 training steps.

**Factorization Accuracy.** In order to understand why our low-rank projection in Trion is more accurate than Power-Iteration in Dion, we compute the $\ell_2-$norm of projection errors between the accumulators $B_t$ and the corresponding orthogonal updates that each optimizer uses to update the model: $\Delta_t^{\text{dion}} = ||B_t - P_t Q_t^\top||_2$ for Dion and $\Delta_t^{\text{trion}} = ||B_t - O_t||_2$ for Trion. In Figure 2 we show the projection errors for the linear layers of the first transformer block in a Llama-30M model and provide more information in Appendix H.

**PT with LDAdamW.** We train Llama-800M on 80B tokens (100 tokens per parameter) from C4 using LDAdam and DCT-AdamW described in Algorithm 2. In particular for our approach, we use EF quantized to 8-bits and some further optimizations from the ZeRO-redundancy optimizer (Rajbhandari et al., 2020), where one layer is updated on a single GPU and then it is broadcasted to the other GPUs. This way, we obtain lower memory usage by replacing redundant operations in the optimizer with communication, since the GPUs receiving the updated layer parameters do not allocate any state. In Figure 3 we present the training loss curves for full-rank AdamW (for reference), LDAdamW and DCT-AdamW and we are interested in directly comparing the last two. We observe that DCT-AdamW has lower training loss than LDAdamW, which also translates to lower perplexities in Table 2. Due to relatively high rank, the memory usage of LDAdamW is close to the memory usage of AdamW because it stores two projection matrices to be able to rotate the momentum buffers. In contrast, DCT-AdamW stores only two sets of $r$ indices instead of storing the actual projection matrices, which drastically reduces the memory usage, coupled with the ZeRO-redundancy trick. In terms of running time, DCT-AdamW is faster than LDAdamW by 10h 7m ($\approx 25.75\%$) and slower than AdamW by 1h 55m ($\approx 5\%$).

**Other PT Ablations.** We replace the SVD projection with our DCT approach and test it on Llama-800M model using 16B tokens from the C4 against FRUGAL/FIRA. For space constraints reasons, we present the results in Appendix I. In addition, we provide some pre-training results for DCT-Adam and SubTrack++ Rajabi et al. (2025) in Appendix J.

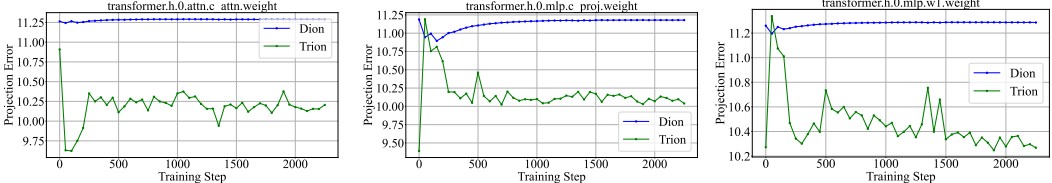

Figure 2: Projection errors for Dion and Trion for a few layers from the first transformer block on Llama-30M ($d = 640, r = 128$).

| | AdamW | LDAdamW | DCT-AdamW |
|---|---|---|---|
| **Train PPL** | 12.88 | 15.10 | 14.95 |
| **Val. PPL** | 11.73 | 13.91 | 13.69 |
| **Mem. (GiB)** | 73.72 | 72.10 | 57.82 |
| **Time** | 1d 13h 22m | 2d 1h 24m | 1d 15h 17m |

Table 2: Pre-training results on Llama-800M for AdamW, LDAdamW and DCT-AdamW with 100 tokens/parameter. AdamW is the full-rank optimizer and is added for reference.

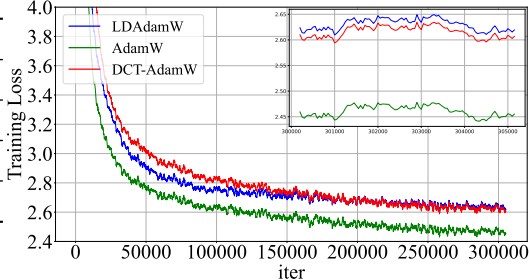

Figure 3: Pre-training Llama-800M with DCT-AdamW and LDAdam on 80B tokens from C4.

## 4 THEORETICAL GUARANTEES

In this section, we provide a theoretical justification for our two-step procedure to approximate SVD-based gradient projections. First, we rigorously show that adaptively selecting columns of an orthogonal matrix based on their alignment with the gradient matrix is an optimal strategy for minimizing the reconstruction error. This approach leads to a contractive compression scheme, which is commonly exploited in the analysis of compressed adaptive optimization algorithms. Next,

to justify the specific use of the DCT matrix, we demonstrate that it naturally serves as a linear approximation of the left or right eigenbases of the gradient matrices.

## 4.1 OPTIMALITY OF NORM-BASED RANKING PROCEDURE

Let $G \in \mathbb{R}^{n \times m}$ be the gradient matrix and $Q \in \mathbb{R}^{n \times n}$ is orthogonal, namely $QQ^\top = I_n$ (without loss of generality, we consider left multiplication). For a given rank $r \leq n$, let $Q_r$ be a $n \times r$ sub-matrix of $Q$ composed of $r$ columns. We are interested in the reconstruction error between $G$ and $Q_r Q_r^\top G$, where $Q_r^\top$ performs projection and $Q_r$ performs back-projection. While we may choose any matrix norm to measure the reconstruction error, the standard Frobenius norm makes the derivation cleaner. Specifically, using the definition of the Frobenius norm $\|G\|_F^2 = \mathrm{tr}(G^\top G)$, linearity of trace and $Q_r^\top Q_r = I_r$ due to orthogonality of $Q$, we decompose the reconstruction error:

$$
\begin{aligned}
\|G - Q_r Q_r^\top G\|_F^2 &= \mathrm{tr}(G^\top (I - Q_r Q_r^\top)^\top (I - Q_r Q_r^\top)G) = \mathrm{tr}(G^\top (I - Q_r Q_r^\top)G) \\
&= \mathrm{tr}(G^\top G - G^\top Q_r Q_r^\top G) = \mathrm{tr}(G^\top G) - \mathrm{tr}(G^\top Q_r Q_r^\top G) \\
&= \|G\|_F^2 - \|Q_r^\top G\|_F^2 = \|G\|_F^2 - \sum_{i=1}^r \|q_i^\top G\|_2^2,
\end{aligned}
$$

where $q_i$'s are the columns of $Q_r$. From this identity, we conclude that to minimize the reconstruction error, the optimal strategy is to maximize the alignments $\|q_i^\top G\|_2^2$ for all selected columns $q_i$. Furthermore, since $\|q_1^\top G\|_2^2 + \|q_2^\top G\|_2^2 + \cdots + \|q_n^\top G\|_2^2 = \|Q^\top G\|_F^2 = \mathrm{tr}(G^\top QQ^\top G) = \mathrm{tr}(G^\top G) = \|G\|_F^2$, following the optimal strategy of selecting $r$ columns from $Q$, we have $\|G - Q_r Q_r^\top G\|_F^2 = \|G\|_F^2 - \sum_{i=1}^r \|q_i^\top G\|_2^2 \leq \|G\|_F^2 - \frac{r}{n} \sum_{i=1}^n \|q_i^\top G\|_2^2 = \left(1 - \frac{r}{n}\right) \|G\|_F^2$. Thus, the proposed norm-based selection strategy for any orthogonal matrix $Q$ induces a low-rank compression scheme that is contractive with a factor $1 - r/n$. Contractivness of the compression scheme is the key property in the convergence analysis of compressed optimization (Stich et al., 2018; Richtárik et al., 2021; Li et al., 2022; Modoranu et al., 2024; Robert et al., 2025).

We can extend the above argument for any $p$-norm over vectorized matrices, for which the Frobenius norm is the special case of $p = 2$. Then, we have:

$$
\|G - Q_r Q_r^\top G\|_p = \left\| \sum_{i=1}^n q_i q_i^\top G - \sum_{i=1}^r q_i q_i^\top G \right\|_p = \left\| \sum_{i=r+1}^n q_i q_i^\top G \right\|_p \overset{(a)}{\leq} \sum_{i=r+1}^n \left\| q_i q_i^\top G \right\|_p
$$

$$
\overset{(b)}{=} \sum_{i=r+1}^n \|q_i\|_p \left\| q_i^\top G \right\|_p \overset{(c)}{\leq} \max(1, n^{\frac{1}{p} - \frac{1}{2}}) \sum_{i=r+1}^n \left\| q_i^\top G \right\|_p,
$$

where (a) follows from the triangle inequality of the $p$-norm, (b) follows from the definition of $p$-norms for matrices, namely $\|uv^\top\|_p = (\sum_{i,j} u_i^p v_j^p)^{1/p} = (\sum_i u_i^p \sum_j v_j^p)^{1/p} = \|u\|_p \|v\|_p$ for two column-vectors $u$ and $v$ of the same size, (c) follows from the relationship between $\ell_p$ norms and that $\|q_i\|_2 = 1$, i.e., $\|v\|_p \leq n^{\frac{1}{p} - \frac{1}{2}} \|v\|_2$ if $p \leq 2$ and $\|v\|_p \leq \|v\|_2$ if $p \geq 2$.

## 4.2 DCT AS LINEAR APPROXIMATION OF THE GRADIENT EIGENBASIS

Given a real-valued gradient matrix $G \in \mathbb{R}^{n \times m}$ and its SVD decomposition $G = U\Sigma V^\top$, our goal is to find fast approximation of (without loss of generality) its left eigenvectors stacked in $U \in \mathbb{R}^{n \times n}$. As $GG^\top = U\Sigma^2 U^\top$, it is equivalent to approximate the eigenvectors of symmetric matrix $GG^\top$.

Our argument starts from a linear algebra decomposition result, originally motivated by the optical information processing literature to factorize linear transformations that can be implemented optically (Müller-Quade et al., 1998; Schmid et al., 2000). The result states that any square matrix $M$ of shape $n \times n$ over the complex numbers $\mathbb{C}$ can be decomposed into a product of diagonal and circulant matrices, i.e., $M = D_1 C_2 D_3 \ldots D_{2k-3} C_{2k-2} D_{2k-1}$, where $D$'s are diagonal matrices and $C$'s are circulant matrices. Circulant matrices are a special class of Toeplitz matrices in which each row is a cyclic right shift of the previous one as shown below:

$$
C = \begin{bmatrix} c_0 & c_1 & c_2 & \cdots & c_{n-1} \\ c_{n-1} & c_0 & c_1 & \cdots & c_{n-2} \\ \vdots & \vdots & \cdots & \vdots & \vdots \\ c_1 & c_2 & \cdots & c_{n-1} & c_0 \end{bmatrix}, \quad F = \frac{1}{\sqrt{n}} \left[ w^{ij} \right]_{i,j=0}^{n-1}, \text{ where } w = e^{\frac{2\pi i}{n}}. \quad (1)
$$

The number of factors $2k - 1$ in this decomposition has been shown to be up to $2n - 1$ for almost all matrices (in the sense of Lebesgue measure), and it is further conjectured that a decomposition

with up to $n$ factors is sufficient (Huhtanen & Perämäki, 2015). Since circulant matrices can be diagonalized using the discrete Fourier transform (DFT) matrix $F$ (see equation 1), i.e., $C = F^*DF$[4] (Golub & Van Loan, 2013), we can decompose the matrix $F^*MF$ with $M = GG^\top$ and arrive at the following decomposition for $GG^\top$:

$$GG^\top = (FD_1F^*)D_2(FD_3F^*)D_4 \cdots D_{2k-2}(FD_{2k-1}F^*). \qquad (2)$$

Notice the analogy between this matrix decomposition and the Taylor expansion for functions. Since the space of matrices is finite-dimensional, the signal-matrix can be recovered using finitely many "variable" $D$'s, in contrast to the infinite series required for function expansions. Keeping this analogy in mind, just as loss functions are linearly approximated at each iteration in first-order optimization algorithms, we consider a "linear" approximation of the decomposition equation 2 by including only a single variable $D$, i.e., $GG^\top \approx FD_1F^*$.

This directly implies that we approximate the eigenvectors $U$ by the DFT matrix $F$. However, since the matrix $GG^\top$ is real and symmetric by design, its eigenvalues are real, and the real part $\text{Re}(F)$ also forms an approximate eigenbasis that better aligns with $U$. Finally, we observe that the real part $\text{Re}(F)$ corresponds to the discrete cosine transform (DCT), up to minor variations.

## 5 RELATED WORK

Our approach aims to improve existing optimizers from prior work in the literature, which we group into two categories: optimizers that speed up convergence at the expense of increased FLOPs and optimizers focused on reducing the running time and memory usage by compressing the gradient via low-rank matrix factorization.

**Fast Convergence Optimizers.** Muon optimizer (Jordan et al., 2024) has recently stood up in the literature for its fast convergence rate due to the orthogonalized momentum update. The purpose of the orthogonalization is to push the singular values of the momentum matrix towards 1. It speeds up convergence by increasing the importance (singular values) of directions in the momentum matrix, which otherwise would have a low impact over the optimization.

The most straightforward approach to computing an orthogonal update is to use the $UV^\top$ from the SVD decomposition of the momentum matrix. Since SVD is expensive, it is replaced by an iterative procedure called Newton-Schulz, which involves computing the odd powers of the momentum matrix up to $5^{th}$ order and multiplying by some carefully chosen constants. While the Newton-Schulz procedure delivers an accurate approximation of $UV^\top$, the odd powers in the polynomial involve full-size matrix multiplications. This is in particular difficult for large scale settings, where the full matrices have to be materialized on a GPU before running Newton-Schulz, which increases communication and memory usage. Dion optimizer (Ahn et al., 2025) aims to reduce the communication overhead by using low-rank, orthogonal updates computed via Power-Iteration, that requires QR decomposition with running time depending on the rank.

Instead, we propose to reduce the overhead of Newton-Schulz in Muon and QR-decomposition in Dion by factorizing the momentum to a low-rank matrix using our DCT-based dynamic column selection approach and orthogonalizing the low-rank momentum using Newton-Schulz, then project back to the original space to update the model parameters.

**Low-Rank Compression of Optimizer States.** Most approaches use individual projection matrices tailored to the gradient at each layer by invoking techniques based on quantization or matrix factorization, such as SVD, QR or PCA to reduce the memory usage of the **optimizer states stored in GPU memory**. The most common approach is to factorize the gradient to low-rank matrices. One pioneering work in the context of low-rank adaptive optimization is the recent GaLore optimizer (Zhao et al., 2024), which performs SVD once at a few steps to project the gradient to a lower-dimensional space. GaLore was followed by several other optimizers that improve certain aspects of it. LDAdam (Robert et al., 2025) aims to improve the computational runtime of GaLore by compressing the first order momentum in AdamW, replacing SVD with a Block Power-Iteration (Bentbib & Kanber, 2015) and performing a smooth subspace transition by rotating the first and second momentum accordingly to incorporate gradients from the same subspace at each step.

By default, GaLore discards the projection error, which LDAdam stores and incorporates into the gradient at the next step to improve convergence. In contrast, FRUGAL (Zmushko et al., 2024)

---

[4] $F^*$ is the conjugate transpose of the complex matrix $F$.

makes the distinction between the low-rank gradient (called state-full) and the projection error (called state-free). The state-full gradient is used in AdamW, while the state-free gradient is fed to an optimizer without a state, such as SignSGD. The main idea is to leverage the remaining gradient information in the projection error since it is available at each step instead of discarding or storing it. A concurrent work to FRUGAL is FIRA (Chen et al., 2024), which preserves the low-rank constraint for memory efficiency while achieving full-rank performance by properly scaling the projection error.

Another approach worth mentioning is Online Subspace Descent Liang et al. (2024), which replaces SVD projection with Online PCA and involves computing the projection matrix as a solution of an optimization objective focused on: (a) minimizing the projection error and (b) forcing the projection matrix to be orthogonal. The authors indicate that performing one step with Adam to solve this additional optimization problem is enough to obtain a qualitatively good projection matrix $P$. In contrast, our method does not introduce such overheads during training since the DCT matrix is already computed at the beginning of training.

Despite all aforementioned approaches being similar in using SVD/QR-based projections, they differ in a few aspects. The most important one in our view is how they handle the projection error and how often they update the low-dimensional subspace, which we clarify in Table 3.

Our approach uses DCT projection coupled with a dynamic column selection to determine a projection matrix tailored to the gradient/momentum for a particular layer and can be integrated into any optimizer, regardless of the way it handles the projection error.

## 6 CONCLUSION AND LIMITATIONS

We introduced the Trion and DCT-AdamW optimizers that use our DCT-based dynamic column selection to replace two techniques used to perform low-rank decomposition, that is, the inaccurate Power-Iteration in Dion and the expensive SVD and QR-decomposition in FRUGAL/FIRA/LDAdam. We showed that our work improves running time and memory usage. Moreover, it recovers the accuracy of the original methods and thus serves as a cheaper alternative to these expensive and inaccurate methods used to perform low-rank decomposition in adaptive gradient methods for both pretraining and finetuning.

Our experiments are limited to models with at most 1.3B parameters for pretraining and additional work is required to test our technique for larger models and beyond the Chinchilla-optimal token counts, which would require significantly more computational resources.

| Low-rank Projection | Type | Frequency* | Error |
|---|---|---|---|
| GaLore (Zhao et al., 2024) | SVD | 200 | discard |
| FRUGAL (Zmushko et al., 2024) | SVD, Random, RandPerm | 200 | feed to SignSGD |
| FIRA (Chen et al., 2024) | SVD | 200 | norm-based scaling |
| LDAdam (Robert et al., 2025) | Block Power-Iteration | 1 | error feedback |
| Dion (Ahn et al., 2025) | Power-Iteration | 1 | save to momentum |
| **Trion (this work)** | DCT | 1 | same as Dion |
| **DCT-AdamW (this work)** | DCT | any | error feedback |

Table 3: Properties of prior low-rank adaptive optimizers. The update frequency* 200 is the default in GaLore that made the approach computationally feasible.

## 7 ACKNOWLEDGEMENTS

We would like to thank the Scientific Computing Department at ISTA for providing access to computational resources to develop this work.

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

CONTENTS

## A    EFFICIENT COMPUTATION OF THE DCT MATRIX ON GPU

In this section we present how we can efficiently compute the DCT-III matrix on a GPU using a vectorized implementation that is fast for large values of $n$ on the GPU. To create the DCT-III matrix $Q \in \mathbb{R}^{n \times n}$, we first create one vector $L = [0, \cdots, n-1]^\top$, which is used to create the matrix $\mathcal{I} \in \mathbb{N}^{n \times n}$ by replicating $L$ on the columns of $\mathcal{I}$:

$$
L = \begin{bmatrix} 0 \\ 1 \\ \vdots \\ n-1 \end{bmatrix} \quad
\mathcal{I} = \begin{bmatrix} 0 & \cdots & 0 \\ 1 & \cdots & 1 \\ \vdots & \cdots & \vdots \\ n-1 & \cdots & n-1 \end{bmatrix}_{n \times n} \quad
Q = \sqrt{\frac{2}{n}} \cos\left( \frac{\mathcal{I} \odot (2\mathcal{I}^\top + 1)}{2n} \pi \right)
$$

We restate that we need to divide the first row by $\sqrt{2}$ to make $Q$ orthogonal. The computational efficiency comes from the element-wise product $\mathcal{I} \odot (2\mathcal{I}^\top + 1)$ that actually computes the integer entries $i(2j + 1)$. The DCT-II matrix can be obtained by transposing the DCT-III.

## B    DYNAMIC COLUMN SELECTION APPROACH

In this section we provide further details about our dynamic column selection approach. When we compute the similarities $S$, we look at the columns of $S$ where on $i^{th}$ column we have the scalar products between all rows of $G$ and the $i^{th}$ column of $Q$. By choosing the columns with largest $\ell_1-$ or $\ell_2$-norms we make sure we select the columns with largest overall alignment with all rows in $G$.

$$
S = GQ = \begin{bmatrix} -G_1- \\ -G_2- \\ \vdots \\ -G_n- \end{bmatrix} \begin{bmatrix} | & | & \cdots & | \\ Q_1 & Q_2 & \cdots & Q_n \\ | & | & \cdots & | \end{bmatrix} = \begin{bmatrix} G_1^\top Q_1 & G_1^\top Q_2 & \cdots & G_1^\top Q_n \\ G_2^\top Q_1 & G_2^\top Q_2 & \cdots & G_2^\top Q_n \\ \vdots & \vdots & \cdots & \vdots \\ G_n^\top Q_1 & G_n^\top Q_2 & \cdots & G_n^\top Q_n \end{bmatrix} \tag{3}
$$

## C    MOTIVATION OF DISCRETE COSINE TRANSFORM MATRIX

In general, any orthogonal matrix would yield similar quantitative results when used with our dynamic column selection technique. In this section we are interested in exploring different types of orthogonal matrices and our goal is to find a candidate matrix whose structure allows us to reduce the computational complexity for the operation $S = GQ$, especially for large layers.

The first candidate is a random orthogonal matrix $Q$, which can be obtained by generating a random Gaussian matrix and then orthogonalizing it using QR-decomposition (e.g. keeping only the Q-component from the decomposition), which should be done only once at the beginning of training. While this would work, it has the drawback of always having $O(n^3)$ complexity when computing the similarities $S$ because the matrix does not have any structure that would allow us to use an algorithm with lower complexity.

The second candidate is the Hadamard matrix, a particular Fourier matrix containing only values $\pm 1$, which is used in the model compression literature and is known for its fast multiplication routines tailored to GPUs. However, it has the drawback of being ill defined for certain values of model's dimension $d_{\text{model}}$ and the current existing procedures provide a matrix that is not orthogonal in these cases, making it unusable for our context.

Our preferred candidate is the DCT matrix, which is also a Fourier matrix and has a particular structure that gives it the potential of computing $S = GQ$ ($S$ is called the discrete cosine transform of $G$), making it faster than the $O(n^3)$ complexity of matmul, as we present below.

When $Q$ is the DCT-II matrix, we can reduce the complexity of $S = GQ$ from $O(n^3)$ to $O(n^2 log(n))$ by using the Makhoul's $N$-point algorithm Makhoul (1980) to compute DCT faster. We summarize the algorithm in Appendix D and also provide a benchmark for different layer sizes and data types encountered in practice. In short, this is a FFT-based algorithm that benefits from the pre- and post-processing steps of the input matrix $G$ (such as permutations and multiplications with complex values) to reduce the computational complexity. Interestingly, the link between Makhoul's algorithm to compute the cosine transform and the same transformation computed via a basic matmul is that the

matmul version embeds all operations of the Makhoul's algorithm in the DCT-II matrix itself, but at a higher computational cost.

Given the current data types implemented in PyTorch, Makhoul's algorithm can be run only on float32 inputs, which makes it infeasible for practical bfloat16 inputs unless the input size is a power of 2. The limitation comes from the lack of complex-bfloat16 type support in PyTorch to represent the real and imaginary parts of a complex number as bfloat16. See Appendix D for more details.

## D  MAKHOUL'S ALGORITHM EXPLAINED

Makhoul's $N$-point algorithm to compute a fast DCT can be summarized as follows:

1. apply a permutation to the input signal $X$ to obtain $X_{PERM}$: vector $[a, b, c, d, e, f]$ becomes $[a, c, e, f, d, b]$, where odd indices are in increasing order, even indices are in decreasing order and they are interleaved (can be cached for the same input size);

2. compute FFT of the permuted signal $X_{PERM}$ and obtain $X_{FFT}$;

3. compute Fourier coefficients $W_k = exp(-2i\pi k/N)$ (can be cached for the same input size);

4. obtain DCT of $X$ as $X_{DCT} = Real(X_{FFT} * W)$ (multiplication by $W$ accounts for permuting the input $X$).

Makhoul's algorithm performs a one-dimensional, type-II DCT for each row of matrix $G$. In essence, the Makhoul's algorithm $S = \text{MAKHOUL}(G)$ is equivalent to the matmul version $S = GQ$, where the matrix $Q$ embeds all the operations performed by the Makhoul's procedure.

We benchmark Makhoul's algorithm against the matmul version on GPU for different layer sizes and show it is faster than matmul for large layers. For small layers (up to embedding size 2048, the benefits of Makhoul's algorithm do not have practical benefits compared to matmul - they have similar runtimes). Concretely, we run 10 iterations of warmup before starting to measure the time for 100 runs on the GPU for both matmul version and Makhoul's version, where $G$ is initialized with random gaussian data in float32 and stays fixed during the benchmark.

In Table 4 we show our benchmarking results where gradient matrix $G$ is in float32 format, compare against the DCT transform obtained via matmul by simply employing $S = GQ$ and compare against the Makhoul's N-point algorithm. The column **Ratio** shows how much faster the Makhoul's implementation is compared to the standard matmul. We can get up to $50\times$ speedup for matrices with more columns than rows.

| Input size | Source Model | Matmul time (s) | Makhoul time (s) | Ratio (@ / FFT) |
|---|---|---|---|---|
| (4096, 4096) | Llama-2-7B | 0.00267808 | 0.00033124 | 8.09× |
| (25600, 5120) | Qwen3-32B | 0.02746414 | 0.00259182 | 10.60× |
| (5120, 25600) | Qwen3-32B | 0.13069152 | 0.00262571 | 49.77× |

Table 4: Comparison of Matmul vs. Makhoul runtimes across different input sizes and models for float32. Ratio measures how much faster the Makhoul's algorithm is compared to the standard matmul. Matmul is faster for ratio $< 1$ and Makhoul is faster when ratio $> 1$.

| Input size | Source Model | Matmul time (s) | Makhoul time (s) | Ratio (@ / FFT) |
|---|---|---|---|---|
| (4096, 4096) | Llama-2-7B | 0.00018977 | 0.00034137 | 0.56x |
| (25600, 5120) | Qwen3-32B | 0.00184539 | 0.00268176 | 0.69x |
| (5120, 25600) | Qwen3-32B | 0.00968907 | 0.00273731 | 3.54x |

Table 5: Comparison of Matmul (bfloat16) vs. Makhoul (float32) timings across different input sizes and models with different data types.

While we can get a significant speedup for the case $R < C$ for float32, the real practical settings use bfloat16 instead of float32 and for this reason we also run our benchmark for bfloat16 as follows: we generate a random matrix $G$ in float32 which will be used for Makhoul's algorithm and the same

matrix $G$ will be converted to bfloat16. Moreover, the DCT matrix $Q$ will also be stored in bfloat16 such that both operands in $S = GQ$ are in bfloat16. On the other hand, since Makhoul's algorithm uses FFT from PyTorch, we are currently restricted to using float32 inputs because at the moment of developing this work PyTorch does not have the complex-bfloat16 type, where both real and imaginary parts are stored in bfloat16 (half precision is supported only for inputs with sizes power of two). As a result, we run Makhoul's algorithm only in float32 and compare with matmul in bfloat16.

In Table 5 we show our benchmarking results where we compare matmul run with bfloat16 inputs against Makhoul's algorithm run on float32 input. For $R \geq C$, matmul-bfloat16 is consistently faster than Makhoul's algorithm. This is expected because bfloat16 type has higher throughput on GPUs than float32. However, we see that Makhoul's algorithm is $3.5\times$ faster than matmul-bfloat16 for $R < C$, which is a much lower speedup compared to the results in Table 4 for float32.

The FFT-based algorithm shows reduction in the running time in our benchmark, as the theory predicts (e.g. $O(n^2 log(n))$ compared to $O(n^3)$) for float32, while the gains are limited when we compare against matmul-bfloat16. In order to benefit from the faster computation of Makhoul's algorithm, we would need to convert bfloat16 matrices to float32, run the faster procedure, then convert back to bfloat16, which is not feasible because of the additional memory and computational overhead. However, when running training in mixed precision, the gradients are computed in bfloat16 and accumulated in a float32 buffer for precision reasons. Fortunately, we have access to this float32 buffer in the optimizer's step function in PyTorch.

## E    BENCHMARKING MAKHOUL'S ALGORITHM ON DIFFERENT HARWDARE

The primary speed up comes from our Dynamic Column Selection algorithm that replaces SVD/QR, not from the FFT-based algorithm, as all results were obtained using matmul to compute the similarities $S = GQ$ instead of Makhoul's algorithm.

Given two matrices $A$, $B$ with shapes $(m, n)$ and $(n, p)$, the complexity of matrix multiplication $A \cdot B$ is $O(mnp)$. Modern GPUs have powerful tensor-cores specialized in matrix-multiply-accumulate (MMA) operations such as $D = A \times B + C$, which reduce the practical complexity by parallel processing. Of course, the number of multiplications is still of order $O(mnp)$, but they are performed in parallel and therefore the effective running time is lower. The H100 and B200 GPUs from Nvidia have powerful tensor-cores that erase the theoretical benefit of Makhoul's algorithm.

Our benchmark in Appendix D shows the improvements of Makhoul's algorithm on an Nvidia H100 GPU with tensor-cores support for matmul turned off by setting `torch.backends.cuda.matmul.allow_tf32=False`. When this flag is set to True, the matmul is faster than Makhoul's algorithm, as we can see below for a layer of shape $(4096, 4096)$. However, on older GPUs such as RTX 2080, we can see the benefit of Makhoul's algorithm as setting the flag `allow_tf32=True/False` does not have a significant impact over running time for computing the similarities $S$ via matmul or Makhoul.

Our Dynamic Column Selection algorithm can be used with any fixed orthogonal matrix because it minimizes the projection error regardless of the chosen matrix. In particular, the DCT matrix has the benefit of theoretical speedup, which can also be achieved in practice on GPUs with less powerful tensor-cores.

In Table 6 below we benchmark the operation of computing the similarities $S = GQ$ computed via matmul and/or Makhoul's algorithm for some practical matrix sizes that can be found in a standard Llama block with embedding size 4096. We can see that Makhoul is always faster than matmul when the tensor-cores are turned off.

For RTX-2080/3090, using Makhoul's algorithm is indeed more beneficial.

**Important mention**. We would like to emphasize one observation in the following table. Looking at a specific layer such as $(4096, 4096)$, an user owning an Nvidia RTX 2080 GPU can compute the similarities $S$ in 1.3 milliseconds using Makhoul (rows 11-12), which is equivalent to approximately 4 times slower than a matmul on H100 with tensor-cores ON (row 7). This is a great achievement since using Makhoul on RTX-2080 (row 12) would be 8 times faster than applying matmul (row 11).

## F    BENCHMARKING RUNNING TIME OF OPTIMIZER STEP

We created a custom model with only one transformer block where the embedding dimensionality is 4096 and number of heads is 32, resulting in a model with 464M parameters in total (embeddings,

Table 6: Matmul (@) vs. Makhoul (FFT) runtime comparison across GPUs and matrix sizes

| Matrix Size | GPU | Tensor-Cores | Row # | Matmul (ms) | FFT (ms) | Ratio | Faster |
|---|---|---|---|---|---|---|---|
| (12288, 4096) | H100 | ✓ | 1 | 1.31 ms | 1.07 ms | 1.21 × | FFT |
| | | ✗ | 2 | 8.07 ms | 1.06 ms | 7.58 × | FFT |
| | RTX 3090 | ✓ | 3 | 11.07 ms | 2.62 ms | 4.22 × | FFT |
| | | ✗ | 4 | 16.46 ms | 2.62 ms | 6.28 × | FFT |
| | RTX 2080 | ✓ | 5 | 30.51 ms | 3.88 ms | 7.85 × | FFT |
| | | ✗ | 6 | 30.79 ms | 3.89 ms | 7.91 × | FFT |
| (4096, 4096) | H100 | ✓ | 7 | 0.36 ms | 0.39 ms | 0.92 × | @ |
| | | ✗ | 8 | 2.67 ms | 0.37 ms | 7.08 × | FFT |
| | RTX 3090 | ✓ | 9 | 4.00 ms | 0.89 ms | 4.48 × | FFT |
| | | ✗ | 10 | 5.84 ms | 0.91 ms | 6.42 × | FFT |
| | RTX 2080 | ✓ | 11 | 10.67 ms | 1.33 ms | 7.99 × | FFT |
| | | ✗ | 12 | 10.81 ms | 1.35 ms | 7.95 × | FFT |
| (11008, 4096) | H100 | ✓ | 13 | 1.19 ms | 0.99 ms | 1.21 × | FFT |
| | | ✗ | 14 | 7.40 ms | 0.95 ms | 7.72 × | FFT |
| | RTX 3090 | ✓ | 15 | 9.93 ms | 2.35 ms | 4.22 × | FFT |
| | | ✗ | 16 | 14.89 ms | 2.37 ms | 6.28 × | FFT |
| | RTX 2080 | ✓ | 17 | 27.62 ms | 3.48 ms | 7.93 × | FFT |
| | | ✗ | 18 | 27.96 ms | 3.49 ms | 8.01 × | FFT |
| (4096, 11008) | H100 | ✓ | 19 | 1.20 ms | 1.41 ms | 0.85 × | @ |
| | | ✗ | 20 | 7.34 ms | 1.41 ms | 5.20 × | FFT |
| | RTX 3090 | ✓ | 21 | 10.03 ms | 3.57 ms | 2.81 × | FFT |
| | | ✗ | 22 | 14.86 ms | 3.59 ms | 4.13 × | FFT |
| | RTX 2080 | ✓ | 23 | 27.83 ms | 5.68 ms | 4.89 × | FFT |
| | | ✗ | 24 | 28.20 ms | 5.69 ms | 4.95 × | FFT |

transformer block, lm-head). We then train this model on a single GPU with sequence length 256 and batch size 1 to minimize the time spent in forward and backward, as well as the memory usage.

Second, we used the model with 800M parameters from Table 1 in our paper and trained it on one GPU with sequence length 512 and batch size 1 to replicate the previous setup.

For both scenarios we measure and report the running time of the optimizer step. This way, we want to show the overhead of low-rank variants with respect to standard, full-rank Muon.

We test full-rank Muon and low-rank optimizers Dion and Trion (with matmul and Makhoul's algorithm) with rank 512 and 1024 to show the overhead of each setting. We perform experiments on one GPU of type Nvidia H100 and Nvidia RTX 3090 because H100 has fast tensor cores that make matmul faster, while RTX 3090 has slower tensor cores that benefit Makhoul's algorithm.

Below we present the results for this synthetic benchmark that emphasizes the overhead of the optimizer step for Trion and Dion in comparison to full-rank Muon:

- in Table 7 we show the benchmarking results for Llama-800M on H100. We observe that matmul is faster than Makhoul's algorithm for both ranks, which is expected because H100 has powerful tensor cores. Trion achieves lower running time than the full-rank Muon, while Dion is very expensive because of the QR-decomposition and our Dynamic Column Selection successfully replaces it.
- in Table 8 we show the benchmarking results for Llama-800M on RTX-3090. We observe that matmul is slower than Makhoul's algorithm for both ranks, which is explained by the slower tensor-cores for this older GPU. Trion achieves lower running time than full-rank Muon for both ranks.
- in Table 9 we show the benchmarking results for our custom Llama-464M on H100. As in the previous case, matmul is faster on H100 than Makhoul's algorithm. Since the layers are larger, we

observe some increase in the overhead for low-rank methods. Still, Trion is much cheaper than Dion, whose running time increases significantly with the rank.

• in Table 10 we benchmark our custom Llama-464M on RTX-3090. Similarly, Makhoul is slightly faster than matmul and the low-rank methods have an additional overhead compared to Muon. Still, Dion is the most expensive also in this case, which is justified by the calls to QR-decomposition.

Table 7: [H100] Llama-800M ($d = 2048$, layers=16, batch-size=1, seq_len=512)

| Method | Rank | Optimizer Step | Memory | Time Overhead | Memory Overhead |
|---|---|---|---|---|---|
| Muon | full | 113 ms/it | 19.07 GB | $1\times$ | $1\times$ |
| Trion+matmul | 512 | 67 ms/it | 18.61 GB | $0.59\times$ | $0.98\times$ |
| Trion+Makhoul | 512 | 78 ms/it | 19.02 GB | $0.69\times$ | $1\times$ |
| Dion | 512 | 246 ms/it | 20.47 GB | $2.18\times$ | $1.07\times$ |
| Trion+matmul | 1024 | 90 ms/it | 18.87 GB | $0.80\times$ | $0.99\times$ |
| Trion+Makhoul | 1024 | 101 ms/it | 19.02 GB | $0.89\times$ | $1\times$ |
| Dion | 1024 | 570 ms/it | 20.65 GB | $5.04\times$ | $1.08\times$ |

Table 8: [RTX-3090] Llama-800M ($d = 2048$, layers=16, batch-size=1, seq_len=512)

| Method | Rank | Optimizer Step | Memory | Time Overhead | Memory Overhead |
|---|---|---|---|---|---|
| Muon | full | 830 ms/it | 18.46 GB | $1\times$ | $1\times$ |
| Trion+matmul | 512 | 355 ms/it | 19.06 GB | $0.43\times$ | $1.03\times$ |
| Trion+Makhoul | 512 | 315 ms/it | 18.66 GB | $0.38\times$ | $1.01\times$ |
| Dion | 512 | 485 ms/it | 19.59 GB | $0.58\times$ | $1.06\times$ |
| Trion+matmul | 1024 | 551 ms/it | 19.06 GB | $0.66\times$ | $1.03\times$ |
| Trion+Makhoul | 1024 | 507 ms/it | 18.66 GB | $0.61\times$ | $1.01\times$ |
| Dion | 1024 | 1010 ms/it | 19.84 GB | $1.22\times$ | $1.07\times$ |

Table 9: [H100] Llama-464M ($d = 4096$, layers=1, batch-size=1, seq_len=256)

| Method | Rank | Optimizer Step | Memory | Time Overhead | Memory Overhead |
|---|---|---|---|---|---|
| Muon | full | 3.19 ms/it | 13.01 GB | $1\times$ | $1\times$ |
| Trion+matmul | 512 | 4.07 ms/it | 13.03 GB | $1.28\times$ | $1\times$ |
| Trion+Makhoul | 512 | 4.41 ms/it | 13.48 GB | $1.38\times$ | $1.04\times$ |
| Dion | 512 | 11.95 ms/it | 15.36 GB | $3.75\times$ | $1.18\times$ |
| Trion+matmul | 1024 | 4.02 ms/it | 13.03 GB | $1.26\times$ | $1\times$ |
| Trion+Makhoul | 1024 | 4.45 ms/it | 13.48 GB | $1.39\times$ | $1.04\times$ |
| Dion | 1024 | 22.60 ms/it | 15.63 GB | $7.08\times$ | $1.2\times$ |

## G    PSEUDOCODE OF DCT-ADAMW OPTIMIZER

In this section we present the pseudocode of the DCT-AdamW optimizer, which uses our DCT-based dynamic column selection approach to create a dynamic, low-rank projection matrix use to factorize the gradient to a low-rank matrix.

Following LDAdamW (Robert et al., 2025), DCT-AdamW incorporates low-rank gradients from different subspaces and this requires rotating the momentum buffers using a rotation matrix $R$, where the second momentum buffer $v_t$ is updated using a simpler rule compared to LDAdamW. DCT-AdamW also supports error feedback and it saves only the error induced by the low-rank compression.

Table 10: [RTX-3090] Llama-464M ($d = 4096$, layers=1, batch-size=1, seq_len=256)

| Method | Rank | Optimizer Step | Memory | Time Overhead | Memory Overhead |
|---|---|---|---|---|---|
| Muon | full | 6.70 ms/it | 12.98 GB | $1\times$ | $1\times$ |
| Trion+matmul | 512 | 8.01 ms/it | 12.37 GB | $1.2\times$ | $0.95\times$ |
| Trion+Makhoul | 512 | 7.89 ms/it | 12.95 GB | $1.18\times$ | $1\times$ |
| Dion | 512 | 13.26 ms/it | 14.58 GB | $1.98\times$ | $1.12\times$ |
| Trion+matmul | 1024 | 7.01 ms/it | 12.37 GB | $1.05\times$ | $0.95\times$ |
| Trion+Makhoul | 1024 | 5.78 ms/it | 12.95 GB | $0.86\times$ | $1\times$ |
| Dion | 1024 | 57.29 ms/it | 14.82 GB | $8.55\times$ | $1.14\times$ |

The sets $\mathcal{I}_{prev/crt}$ hold the indices of the $r$ columns for the previous/current projections and $Q_{prev/crt}$ contain the columns from $Q$ specified by the these two sets. $T_u$ represents the subspace update interval (set to 200 for GaLore and to 1 for LDAdam). The procedure RANKCOLS ranks the columns of $S$, which is computed as $S = $ MAKHOUL$(G)$ (see Appendix D) or $S = G_t \cdot Q$ as explained in Section 2.1. In order to make sure the momentum buffers integrate gradients from the same subspaces, we need to rotate $m_t$ and $v_t$ using a rotation matrix $R$ that first projects the momentum to the full-dimensional space using the previous projector and then projects it back to the current lower-dimensional subspace. We can perform this rotation directly in the $r$-dimensional space by multiplying the two projection matrices, resulting in $R \in \mathbb{R}^{r \times r}$. Rotating $v_t$ might introduce negative values and we apply the absolute value function to force the non-negativity of $v_t$. The low-rank gradient $g_t$ is computed by multiplying the full-rank gradient $G_t$ with $Q_{crt}$ and the full-rank update is obtained multiplying $u_t$ by $Q_{crt}^\top$.

---

**Algorithm 2** DCT-AdamW (right projection)

1: Input: $\beta_1, \beta_2, \epsilon, T, T_u, r$
2: $m_0, v_0 \leftarrow 0_{n \times r}, 0_{n \times r}$
3: $\Xi_1 \leftarrow 0_{n \times m}$    ◇ error feedback (EF) buffer
4: $Q \in \mathbb{R}^{m \times m}$    ◇ DCT matrix
5: $\mathcal{I}_{crt}, \mathcal{I}_{prev} \leftarrow 0_r, 0_r$    ◇ column indices
6: **for** $t = \{1, 2, \ldots, T\}$ **do**
7:    $G_t \leftarrow \nabla_\theta f(\theta_t) + \Xi_t$
8:    $R \leftarrow$ UPDATESUBSPACE$(G_t)$
9:    $g_t \leftarrow G_t \cdot Q_{crt}$    ◇ projected gradient
10:    $\Xi_t \leftarrow G_t - g_t \cdot Q_{crt}^\top$    ◇ update EF
11:    $m_t \leftarrow \beta_1 \cdot m_{t-1} \cdot R + (1 - \beta_1)g_t$
12:    $v_t \leftarrow \beta_2 |v_{t-1} \cdot R| + (1 - \beta_2)g_t^2$
13:    $\theta_{t+1} \leftarrow \theta_t - \eta_t \frac{\hat{m}_t}{\epsilon + \sqrt{\hat{v}_t}} Q_{crt}^\top$
14: **end for**

**Algorithm 3** Update Subspace procedure

1: **procedure** UPDATESUBSPACE$(G)$
2:    $R \leftarrow I_{r \times r}$
3:    **if** $t > 1$ **then**
4:      $\mathcal{I}_{prev} \leftarrow \mathcal{I}_{crt}$
5:    **end if**
6:    **if** $(t = 1) \vee (t \mod T_u = 0)$ **then**
7:      $S = $ MAKHOUL$(G)$    ▷ or $S = GQ$
8:      $\mathcal{I}_{crt} \leftarrow$ RANKCOLS$(GQ, r)$
9:      $R \leftarrow Q_{prev}^\top \cdot Q_{crt}$
10:    **end if**
11:    **return** $R$
12: **end procedure**

## H  PROJECTION ERRORS OF TRION AND DION

We train two Llama models with 30M parameters ($d_{\text{model}} = 640$) with rank $r = 128$ using Dion and Trion optimizers and save the gradients. For each set of gradients we simulate the operations of the other optimizer and record the metrics $\Delta_t^{\text{dion}}$ and $\Delta_t^{\text{trion}}$. Our experiments showed no significant differences in the patterns for the two simulations and we plot the projection errors for the simulation that used gradients from the model trained with Dion optimizer. In Figure 2 we show the projection errors for the linear layers in the first transformer block, where Trion yields lower projection error than Dion and the trend is the same across all transformer blocks in the model. These plots explain why Trion has lower training and validation perplexity, as can be seen in Table 1. Moreover, the projection error seems to be constant for Dion, while for Trion the projection error has a decreasing trend for some layers, which once again shows the dynamic behavior of our column selection approach coupled with Newton-Schulz iteration.

# I   PRE-TRAINING WITH FRUGAL/FIRA

**Pre-training with FRUGAL.** We integrate the DCT matrix into the FRUGAL optimizer and compare against the SVD, RandPerm and Random projections. RandPerm uses a random permutation as a projection matrix, while Random generates a random, semi-orthogonal matrix. In Figure 4a we show the training loss for all these runs. In the zoomed-in plot for the last 5000 training steps we see that SVD projection recovers the performance full-rank AdamW, while the DCT projection is a good approximation of the SVD, which supports our claim. We present our numerical results in Table 11. We would like to emphasize the running time reduced by 1h 48m ($\approx 22.6\%$) compared to the SVD projection, as well as the memory usage reduced by about 2.2GB ($\approx 3.5\%$), while the increase in perplexity is less than one point. In comparison to RandPerm and Random projections, the runtime is on par, while train and validation perplexities are lower by approximately one point in the favor of DCT, illustrating again the benefit of our approach.

**Pre-training with FIRA.** We integrate DCT into FIRA optimizer and compare against SVD. In Figure 4b we show the training loss, where our focus is on the comparison between SVD and DCT projections. We observe that DCT consistently yields lower training loss compared to the SVD projection, which is also visible in the numerical results in Table 11, translated to lower perplexities in the favor of DCT. Moreover, the memory usage is smaller by 2GB ($\approx 3\%$) and the running time is lower by 1h 54m ($\approx 23.8\%$).

Table 11: Pre-training Results for AdamW, FRUGAL and FIRA with different projections using 20 tokens/parameter. DCT is a good approximation to SVD with lower runtime and memory. AdamW is the full-rank optimizer and is added for reference.

|  | AdamW | FRUGAL | | | | FIRA | |
|---|---|---|---|---|---|---|---|
|  |  | SVD | DCT | RandPerm | Random | SVD | DCT |
| **Train PPL** | 15.55 | 15.35 | 15.63 | 16.52 | 17.02 | 19.37 | 18.88 |
| **Val. PPL** | 14.05 | 14.02 | 14.23 | 15.18 | 15.61 | 17.67 | 17.30 |
| **Mem. (GiB)** | 73.49 | 65.70 | 63.50 | 65.44 | 63.72 | 68.44 | 66.48 |
| **Time** | 7h 54m | 9h 45m | 7h 57m | 7h 56m | 7h 56m | 9h 53m | 7h 59m |

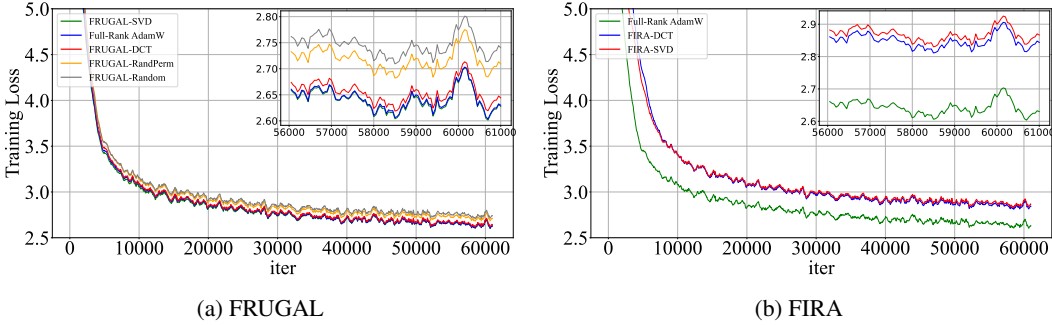

(a) FRUGAL  (b) FIRA

Figure 4: Pre-training Llama-800M using FRUGAL and FIRA on 16B tokens from C4.

# J   PRE-TRAINING WITH DCT-ADAMW AND SUBTRACK++

In this section we provide additional ablations.

**DCT-AdamW vs SubTrack++ Rajabi et al. (2025).** In Table 12 we provide some comparison of DCT-AdamW and SubTrack++ on a model with 350M parameters, rank 256 (25% of model's embedding size) and preconditioner update frequency 200. We run three experiments with different random seeds for each optimizer and report the average metrics. We observe that our DCT-AdamW achieves significantly lower values for train and validation metrics with slightly faster runtime and memory usage, which proves that our Dynamic Column Selection approach is effective in tracking the low-rank subspace of the gradient.

Table 12: Comparison between DCT-AdamW and SubTrack++

| Metric | DCT-AdamW | SubTrack++ |
|---|---|---|
| Train Loss | 2.738 | 2.779 |
| Train PPL | 15.48 | 16.13 |
| Val Loss | 2.747 | 2.789 |
| Val PPL | 15.60 | 16.266 |
| Runtime | 6h 57m | 7h 15m |
| Memory Usage (per GPU) | 42.99 GB | 46.95 GB |

## K  FINE-TUNING

We fine-tune Llama-2 7B on GSM-8k dataset using SVD and DCT projections for FRUGAL and FIRA optimizers, as well as LD-AdamW and DCT-AdamW. We show our results for ranks $r \in \{32, 512\}$ in Table 13. We also finetune Qwen-2.5-7B on GSM-8k using DCT-AdamW, GaLore and AdamW for reference and show the results in Table 14 for the same ranks.

**Fine-tuning with FRUGAL.** Both projections yield similar training loss for both rank values, which is an indication that DCT is indeed a good approximation for SVD. Despite having different training loss, the SVD projection achieves the same accuracy for both ranks. It is surprising that DCT yields better accuracy for lower rank compared to higher rank. In terms of memory usage, as expected for this large model, the DCT projection saves 8GB of memory ($\approx 23.4\%$) for $r = 32$, while for $r = 512$ the reduction in memory is only 6.3GB ($\approx 18.2\%$). The running time is reduced by roughly 35m ($\approx 75\%$) for both ranks.

**Fine-tuning with FIRA.** The DCT projection yields larger training loss compared to SVD. DCT recovers the accuracy, outperforms the SVD for large rank, and on average reduces the memory usage by $\approx 1.35$ GB and the running by $\approx 10$m.

**Fine-tuning with LDAdamW & DCT-AdamW.** In this setting we compare the DCT projection with the block power-iteration used in LDAdamW as an approximation to SVD, both without EF. The training loss achieved is smaller for DCT on $r = 32$ and larger for $r = 512$ compared to LDAdamW. LDAdamW achieves more than 1% accuracy for $r = 32$ and comparable accuracy for $r = 512$. Regarding memory, DCT-AdamW uses $\approx 1$ GB less memory, while achieving a speedup of about 20m for $r = 512$. We would like to emphasize that EF does not help DCT-AdamW in comparison to LDAdamW for $r = 32$, the accuracy of DCT-AdamW with EF being 29.11% vs 32.53% for LDAdam with EF for $r = 32$, while for $r = 512$ the accuracy can be recovered, scoring 35.33% compared to 35.86% for LDAdamW with EF.

**Fine-tuning Qwen-2.5 7B.** In this setting we compare the DCT projection with the original GaLore with our DCT-AdamW without EF, where both optimizers update the subspace once at 200 steps. In Table 14 shows that despite having slightly higher training loss, the DCT-AdamW has higher evaluation accuracy than GaLore.

Table 13: Fine-tuning results for Llama-2 7B on the GSM-8K dataset.

| | FRUGAL | | | | FIRA | | | | LD/DCT-AdamW | | | |
|---|---|---|---|---|---|---|---|---|---|---|---|---|
| | rank 32 | | rank 512 | | rank 32 | | rank 512 | | rank 32 | | rank 512 | |
| | SVD | DCT | SVD | DCT | SVD | DCT | SVD | DCT | LD | DCT | LD | DCT |
| **Train Loss** | 0.046 | 0.059 | 0.051 | 0.071 | 0.123 | 0.209 | 0.094 | 0.192 | 0.261 | 0.176 | 0.101 | 0.208 |
| **Acc. (%)** | 33.81 | 35.93 | 33.81 | 34.26 | 32.15 | 32.45 | 34.27 | 35.25 | 31.38 | 30.09 | 35.61 | 35.17 |
| **Mem. (GB)** | 39.61 | 31.59 | 40.68 | 34.41 | 32.72 | 31.29 | 35.41 | 34.11 | 32.08 | 31.17 | 35.58 | 34.35 |
| **Running Time** | 1h 22m | 46m | 1h 19m | 47m | 1h 8m | 1h | 1h 9m | 1h | 55m | 48m | 1h 8m | 48m |

Table 14: Fine-tuning results for Qwen-2.5-7B on AdamW, DCT-AdamW and GaLore on GSM-8k. DCT is a good approximation to SVD with lower runtime and memory. AdamW is the full-rank optimizer and is added for reference.

| | **AdamW** | **DCT-AdamW** | | **GaLore** | |
|---|---|---|---|---|---|
| | full rank | rank 32 | rank 512 | rank 32 | rank 512 |
| **Train Loss** | 0.012313 | 0.053792 | 0.0545 | 0.0388 | 0.0184 |
| **Eval Acc (%)** | 67.70 | 64.59 | 65.35 | 63.23 | 64.41 |
| **Memory (GB)** | 64.5 | 40.4 | 44 | 40.9 | 45.15 |
| **Running Time** | 48m | 49m | 47m | 58m | 58m |

