# OpenReview forum: "Trion: FFT-based Dynamic Subspace Selection for Low-Rank Adaptive Optimization of LLMs"
_ICLR.cc/2026/Conference — ICLR 2026 Poster_

### Official Review · Reviewer_kABp · 2025-10-29

**Soundness:** 3
**Presentation:** 3
**Contribution:** 3
**Rating:** 8
**Confidence:** 2

**Summary:**

In this paper, the authors have attempted to address the computational and memory overhead in optimizers such as AdamW, Muon, Dion, and GaLore in LLM training. Prior low rank method relies on using SVD or QR decomposition to project gradients into a lower dimensional subspace. However, they are computationally expensive. Therefore, in this paper, the authors attempt to replace these with an alternative low-rank projection approach, which is cheaper to compute.

**Strengths:**

The strengths of this paper are summarized as follows:

1. SVD is time-consuming in LLM training, and it is widely used in various methods, like GaLore, FIRA, and FRUGAL. Replacing SVD with the Fast Fourier Transform based algorithm can reduce the time complexity.

2. Empirically, it has shown improvements on multiple optimizers and model sizes, like LLaMA 350M, 800M, and 1.3B. Also, on all of these model sizes, Trion has shown better performance and better running time compared with Dion.

3. It has a strong theoretical guarantee for justifying the column selection approach may give the most significant column.

**Weaknesses:**

The weaknesses of this paper are summarized as follows:

1. The largest model size in the experiment is 1.3B. It would be better if the authors may consider running an experiment on larger models since modern transformer architectures are getting much larger than this size. Also, only the C4 dataset and LLaMA are tested and there is no fine-tuning or downstream benchmarks.

2. Galore has numerous follow-up works, such as Galore 2 [1], Golore [2], and Sara [3]. Could the authors give a comparison with these works?

[1] DiJia Su, Andrew Gu, Jane Xu, Yuandong Tian, and Jiawei Zhao. "Galore 2: Large-scale llm pre-training by gradient low-rank projection." arXiv preprint arXiv:2504.20437 (2025).

[2] Yutong He, Pengrui Li, Yipeng Hu, Chuyan Chen, and Kun Yuan. "Subspace optimization for large language models with convergence guarantees." ICML'25.

[3] Haochen Zhang, Junze Yin, Guanchu Wang, Zirui Liu, Tianyi Zhang, Anshumali Shrivastava, Lin Yang, and Vladimir Braverman. "Breaking the Frozen Subspace: Importance Sampling for Low-Rank Optimization in LLM Pretraining". NeurIPS'25.

**Questions:**

Please see the weaknesses.

---

> ### Author Response · Authors · 2025-11-21
> **Authors' answers**
>
> We would like to thank the reviewer for their feedback. We address your concerns below.
>
> **Weakness (1)**
>
> Due to space limitations, we presented only the pretraining results in the main body of the paper. The fine-tuning experiments can be found in Appendix H.
>
> Regarding model size, we replicate the setup in Table 1 to train a Llama-7B model with two exceptions:
>
> 1) We use a node with 8x Nvidia B200 GPUs 180GB instead of 8x Nvidia H100 80 GB
>
> 2) We use only 10B tokens (equivalent to 1.5x model size) instead of 20x model size
>
> We are aware that this setup is suboptimal and we hope the reviewers are aware of the costs of training such a model using academic resources.
>
> By analyzing the results provided in the table below, our claim from the paper also holds at this large model scale: Trion exhibits the same running time and memory usage for different ranks. In contrast, the running time and memory usage of Dion depend on the rank because of the QR-decomposition procedure, which we successfully replaced in Trion with our cheaper Dynamic Columns Selection approach.
>
> | Metric | Trion $r=512$ | Trion $r=1024$ | Dion $r=512$ | Dion $r=1024$ |
> |---|---|---|---|---|
> | train loss             | 2.361    | 2.355    | 2.365     | 2.359     |
> | train ppl              | 10.61    | 10.55    | 10.65     | 10.59     |
> | final val loss         | 2.371    | 2.363    | 2.377     | 2.367     |
> | final val ppl          | 10.71    | 10.63    | 10.78     | 10.67     |
> | runtime                | 15h 53m  | 15h 56m  | 17h 6m    | 18h 17m   |
> | memory usage (per GPU) | 149.9 GB | 149.9 GB | 173.87 GB | 175.52 GB |
> |
>
> **Weakness (2)**
>
> In our work we chose to compare against two follow up works on GaLore, such as Frugal and Fira and the results can be seen in Appendix G. Our limited compute budget forced us to restrict our baselines to Dion, Frugal, Fira and LDAdamW and test across a few model sizes and perform some grid search for the learning rates.
>
> While GoLore is similar to our work, we would like to point out some key algorithmic differences between our DCT-AdamW and GoLore:
>
> 1. **Low-Rank projection:**
> - **GoLore**: samples an orthogonal matrix from the Stiefel manifold (low-rank, orthogonal matrix), whic does not provable minimize the projection error.
> - **DCT-AdamW**: we use Dynamic Column Selection to choose the subspace that provably minimizes the projection error via the Dynamic Column Selection approach.
>
> 2. **Updating first order moment of the gradient (buffer $M_t$ in AdamW):**
> - Both GoLore and DCT-AdamW correctly rotate the momentum buffer to incorporate gradients from new subspaces. This change was not implemented in GaLore
>
> 3. **Updating second order moment of the gradient (buffer $V_t$ in AdamW):**
> - **GoLore**: does not rotate the variance buffer $V_t$ and therefore it incorrectly incorporates squared gradients from different subspaces, similarly to what GaLore does. The closest-to-optimal way of updating $V_t$ is explained in the prior work LDAdamW that introduced the rotations for the momentum buffers for the first time (see page 6, Algorithm 1, line 9 from the LDAdamW paper **[1]**).
>
> - **DCT-AdamW**: we observed the update of the variance buffer $V_t$ in LDAdamW can be simplified to $v_t \gets \beta_2 |v_{t-1} \cdot R_t| + (1-\beta_2) g_t^2$, where $g_t$ is the low-rank gradient and $R_t$ is the rotation matrix to project from the previous subspace to the current one (see Algorithm 2 in Appendix E in our supplementary material). In addition, since the variance buffer $v_t$ must be positive and the rotation can result in negative values, we apply the absolute value function to each entry, which is also done in LDAdamW. This is indeed an approximation to the closer-to-optimal expression developed by authors of LDAdamw and the results in Table 2 and Figure 2 show it works reasonably well.
>
>
> **References:**
>
> **[1]** Robert et al., **LDADAM: ADAPTIVE OPTIMIZATION FROM LOW-DIMENSIONAL GRADIENT STATISTICS**, ICML 2025, link https://openreview.net/forum?id=Zkp1GuHerF

---

> ### Author Response · Authors · 2025-11-26
>
> Dear reviewer,
>
> As we are getting closer to the end of the author-reviewer discussion period, we are kindly asking you to have a look at our answers to your questions and please let us know whether they clarified your concerns.
>
> Respectfully, The authors

---

> > ### Comment · Reviewer_kABp · 2025-11-26
> > **Thank you for your response!**
> >
> > I acknowledge that I have read the responses from the authors. All of my concerns are addressed. I choose to maintain my original rating.

---

### Official Review · Reviewer_F7Ct · 2025-10-31

**Soundness:** 3
**Presentation:** 3
**Contribution:** 2
**Rating:** 6
**Confidence:** 3

**Summary:**

This paper tackles the high compute and memory cost of SVD/QR decompositions in low-rank optimizers  The authors propose replacing these expensive, per layer projections with a "dynamic column selection" from a single, predefined orthogonal basis (the Discrete Cosine Transform, or DCT, matrix) computed once at the start . The method efficiently selects the top $r$ most aligned DCT basis vectors for each layer's gradient, replacing SVD/QR with a fast matrix multiplication (or FFT) and a sorting step. This saves memory, as each layer only stores $r$ indices instead of a full projection matrix. The authors integrate this technique into two new optimizers: Trion (improving Dion) and DCT AdamW (improving AdamW variants).

**Strengths:**

1. The method's runtime is rank independent, whereas SVD/QR based methods like Dion get slower as the rank $r$ increases. This is a significant practical advantage for using larger and more expressive ranks.

2. The core idea is simple yet elegant. The "dynamic column selection" is not just heuristic; the authors prove it is the optimal strategy for minimizing reconstruction error, given a fixed orthogonal basis $Q$. The motivation for using the DCT as that basis is also well justified.

3. Strong Empirical Results:

i. Trion vs. Dion (Table 1): Trion consistently achieves better validation perplexity, lower memory usage (~7-10%), and faster runtimes (up to 18% faster) than its direct baseline, Dion, across all tested model sizes and ranks.

ii. DCT-AdamW vs. LDAdamW: DCT-AdamW achieves better validation perplexity, drastically lower memory, and is significantly faster (~25%).

**Weaknesses:**

1. Theoretical speedup not realized. The paper heavily motivates using DCT by citing the fast $O(n^2 \log n)$ FFT-based algorithm. However, the authors admit that for the model sizes tested (up to $d=2048$), this speedup was not significant, and a standard $O(n^3)$ matmul was used15. The primary speedup comes from replacing SVD/QR, not from the FFT.

2. While DCT-AdamW beats its low-rank competitor LDAdamW, it is still significantly outperformed by full rank AdamW (Val. PPL 13.69 vs. 11.73). This shows the method is a better low rank compromise, but still a compromise.

3. As the authors note, experiments are limited to 1.3B models. The method's true scalability and the benefit of the FFT based algorithm would only be evident on much larger models.

**Questions:**

Please see Weaknesses section

---

> ### Author Response · Authors · 2025-11-21
> **[1] Authors' answer for Weakness 1**
>
> We would like to thank the reviewer for their feedback. We address your concerns below.
>
> **Weakness (1)**
>
> We thank the reviewer for this question. We believe it is important to clarify this and will update our paper accordingly to avoid any confusion. Please find our explanations below.
>
> We completely agree with the observations. Indeed, the primary speed up comes from our Dynamic Column Selection algorithm that replaces SVD/QR, not from the FFT-based algorithm, as all results in our paper were obtained using matmul to compute the similarities $S = GQ$ instead of Makhoul’s algorithm.
>
> Given two matrices $A, B$ with shapes $(m, n)$ and $(n, p)$ , the complexity of matrix multiplication $A \cdot B$ is $O(mnp)$ . Modern GPUs have powerful tensor-cores specialized in matrix-multiply-accumulate (MMA) operations, such as $D = A \times B + C$, which reduce the practical complexity by parallel processing. Of course, the number of multiplications is still of order $O(mnp)$, but they are performed in parallel and therefore the effective running time is lower. The H100 and B200 GPUs from Nvidia have powerful tensor-cores that erase the theoretical benefit of Makhoul’s algorithm.
>
> Our benchmark in Appendix D shows the improvements of Makhoul’s algorithm on an Nvidia H100 GPU with tensor-cores support for matmul turned off by setting **torch.backends.cuda.matmul.allow_tf32=False**. When this flag is set to True, the tensor-cores are used and matmul becomes slightly faster than Makhoul’s algorithm, as we can see below for a layer of shape $(4096, 4096)$ (for example, in line 7 of the table below). However, on older GPUs such as RTX 2080, we can see the benefit of Makhoul’s algorithm as setting the flag **allow_tf32=True/False** does not have a significant impact over running time for computing the similarities $S$ via matmul or Makhoul (lines 5-6 in table below).
>
> As we stated in the paper, our Dynamic Column Selection algorithm can be used with any fixed orthogonal matrix because it minimizes the projection error regardless of the chosen matrix. In particular, the structure of the DCT matrix allows using Makhoul's algorithm that has the benefit of theoretical speedup, which can also be achieved in practice on GPUs with less powerful tensor-cores. We see this as a feature that our approach can benefit from rather than a drawback because the gains of Dynamic Column Selection approach are still valid, with or without the use of Makhoul's algorithm.
>
> In the table below we benchmark the operation of computing the similarities $S = GQ$ computed via matmul and/or Makhoul’s algorithm for some practical matrix sizes that can be found in a standard Llama block with embedding size $4096$. We can see that Makhoul is always faster than matmul when the tensor-cores are turned off.
>
> For RTX3090 and older, using Makhoul’s algorithm is indeed more beneficial, as we showed in our answer for **Weakness #3 of Reviewer #1 ZMSS**.
>
> **Important mention**. We would like to emphasize one observation in the following table. Looking at a specific layer such as $(4096, 4096)$, an user owning an Nvidia RTX 2080 GPU can compute the similarities $S$ in $1.3$ milliseconds using Makhoul (rows 11-12), which is equivalent to approximately 4 times slower than a matmul on H100 with tensor-cores ON (row 1). This is a great achievement since it would be 8 times faster than applying matmul (rows 11-12).
>
> In the table below we use the abbreviation **MM @** for matmul and **FFT** for Makhoul's algorithm to improve visibility.
>
> | # | matrix size | GPU | tensor-cores | MM=matmul (ms) | FFT=Makhoul (ms) | Ratio matmul/FFT | Which is faster? |
> |---|---|---|---|---|---|---|---|
> |  1 | (12288, 4096) | H100 | ON  |  1.31 | 1.07 | 1.21x | FFT |
> |  2 | | | OFF |  8.07 | 1.06 | 7.58x | FFT |
> |  3 | | RTX 3090 | ON  | 11.07 | 2.62 | 4.22x | FFT |
> |  4 | | | OFF | 16.46 | 2.62 | 6.28x | FFT |
> |  5 | | RTX 2080 | ON  | 30.51 | 3.88 | 7.85x | FFT |
> |  6 | |  | OFF | 30.79 | 3.89 | 7.91x | FFT |
> |
> |  7 | (4096, 4096)  | H100 | ON  |  0.36 | 0.39 | 0.92x | MM @ |
> |  8 | | | OFF |  2.67 | 0.37 | 7.08x | FFT |
> |  9 | | RTX 3090 | ON  |  4.00 | 0.89 | 4.48x | FFT |
> | 10 | | | OFF |  5.84 | 0.91 | 6.42x | FFT |
> | 11 | | RTX 2080 | ON  | 10.67 | 1.33 | 7.99x | FFT |
> | 12 | | | OFF | 10.81 | 1.35 | 7.95x | FFT |
> |
> | 13 | (11008, 4096) | H100 | ON  |  1.19 | 0.99 | 1.21x | FFT |
> | 14 | | | OFF |  7.40 | 0.95 | 7.72x | FFT |
> | 15 | | RTX 3090 | ON  |  9.93 | 2.35 | 4.22x | FFT |
> | 16 | | | OFF | 14.89 | 2.37 | 6.28x | FFT |
> | 17 | | RTX 2080 | ON  | 27.62 | 3.48 | 7.93x | FFT |
> | 18 | | | OFF | 27.96 | 3.49 | 8.01x | FFT |
> |
> | 19 | (4096, 11008) | H100 | ON  | 1.20 | 1.41 | 0.85x | MM @ |
> | 20 | | | OFF | 7.34 | 1.41 | 5.20x | FFT |
> | 21 | | RTX 3090 | ON | 10.03 | 3.57 | 2.81x | FFT |
> | 22 | | | OFF | 14.86 | 3.59 | 4.13x | FFT |
> | 23 | | RTX 2080 | ON | 27.83 | 5.68 | 4.89x | FFT |
> | 24 | | | OFF | 28.20 | 5.69 | 4.95x | FFT |
> |

---

> ### Author Response · Authors · 2025-11-21
> **[2] Authors' answer for Weaknesses 2 & 3**
>
> **Weakness (2)**
>
> We agree with the reviewer on this point. In our work we take into consideration the 2D structure of Linear layers in the Transformer blocks and therefore perform low-rank compression of the corresponding 2D matrices. We focus on reducing the overhead of SVD/QR-based techniques applied to the particular area of low-rank compression of optimizer states.
>
> On the other hand, AdamW updates parameters using element-wise operations and does not take into consideration the 2D structure of the gradients. While AdamW outperforms both LDAdamW and DCT-AdamW in Table 2 and Figure 2, our main comparison should be between LDAdamW and DCT-AdamW in terms of memory usage, running time and train/validation perplexities as our work aims to replace the QR-decomposition from LDAdamW.
>
> **Weakness (3)**
>
> We replicate the setup in Table 1 to train a Llama-7B model with two exceptions:
>
> 1) We use a node with 8x Nvidia B200 GPUs 180GB instead of 8x Nvidia H100 80 GB
>
> 2) We use only 10B tokens (equivalent to 1.5x model size) instead of 20x model size
>
> We are aware that this setup is suboptimal and we hope the reviewers are aware of the costs of training such a model using academic resources.
>
> By analyzing the results provided in the table below, our claim from the paper also holds at this large model scale: Trion exhibits the same running time and memory usage for different ranks. In contrast, the running time and memory usage of Dion depend on the rank because of the QR-decomposition procedure, which we successfully replaced in Trion with our cheaper Dynamic Columns Selection approach.
>
> | Metric | Trion $r=512$ | Trion $r=1024$ | Dion $r=512$ | Dion $r=1024$ |
> |---|---|---|---|---|
> | train loss             | 2.361    | 2.355    | 2.365     | 2.359     |
> | train ppl              | 10.61    | 10.55    | 10.65     | 10.59     |
> | final val loss         | 2.371    | 2.363    | 2.377     | 2.367     |
> | final val ppl          | 10.71    | 10.63    | 10.78     | 10.67     |
> | runtime                | 15h 53m  | 15h 56m  | 17h 6m    | 18h 17m   |
> | memory usage (per GPU) | 149.9 GB | 149.9 GB | 173.87 GB | 175.52 GB |
> |

---

> ### Author Response · Authors · 2025-11-26
>
> Dear reviewer,
>
> As we are getting closer to the end of the author-reviewer discussion period, we are kindly asking you to have a look at our answers to your questions and please let us know whether they clarified your concerns.
>
> Respectfully, The authors

---

### Official Review · Reviewer_tJpX · 2025-11-01

**Soundness:** 3
**Presentation:** 3
**Contribution:** 3
**Rating:** 4
**Confidence:** 4

**Summary:**

The authors introduce a Discrete Cosine Transform (DCT)-based dynamic column selection technique that approximates optimal low-rank projections by selecting columns of a fixed orthogonal DCT matrix aligned with each layer’s gradients to replace expensive SVD/QR-based low-rank projections used in adaptive optimizers for large language models. This method reduces computation and memory overhead by avoiding per-layer decompositions and storing only column indices. They apply this idea in two new optimizers: Trion, which improves Dion, and DCT-AdamW, a low-rank variant of AdamW.

**Strengths:**

1. The work targets a bottleneck in large-model training: the cost of SVD/QR-based low-rank projections used in adaptive optimizers.
2. The proposed DCT-based projection method is straightforward and easily integrable into existing optimizers.
3. The approach achieves good efficiency gains.
4. The paper provides mathematical rationale for why DCT approximates the gradient eigenbasis and the effectiveness of norm-based selection.
5. The presentation is easy to follow and the paper is well-written.

**Weaknesses:**

1. All experiments stop at 1.3B-parameter models; there is no validation on pretraining larger LLMs (7B+), which undermines claims of scalability.
2. The paper lacks certain ablations of critical design choices (e.g., norm type, rank sensitivity, DCT variant).
3. Distributed and FSDP discussions are not empirically backed with wall-clock or communication cost benchmarks.
4. While the paper targets efficiency, comparison with fast efficient baselines that are not using SVD/QR decomposition like APOLLO [1] and SubTrack++ [2] would help to strengthen the paper.
---
[1] Zhu et al., 2025. APOLLO: SGD-like Memory, AdamW-level Performance.
[2] Rajabi et al., 2025. SubTrack++: Gradient Subspace Tracking for Scalable LLM Training

**Questions:**

Please refer to the weaknesses.

---

> ### Author Response · Authors · 2025-11-21
> **Authors' answer to Weaknesses 1,2,3,4**
>
> We would like to thank the reviewer for their feedback. We address your concerns below.
>
> **Weakness (1)**
>
> We replicate the setup in Table 1 to train a Llama-7B model, with two exceptions:
> 1) We use a node with 8x Nvidia B200 180GB instead of 8x Nvidia H100 80 GB
> 2) We use only 10B tokens (equivalent to 1.5x model size) instead of 20x model size
>
> We acknowledge that this setup is suboptimal and we hope the reviewers are aware of the costs of training such a model using academic resources. Please find our results in the table below.
>
> By analyzing the results provided in the table below, our claim from the paper also holds at this large scale: Trion exhibits the same running time and memory usage for different ranks. In contrast, the running time and memory usage of Dion depend on the rank because of the QR-decomposition procedure, which we successfully replaced in Trion.
>
> | Metric | Trion $r=512$ | Trion $r=1024$ | Dion $r=512$ | Dion $r=1024$ |
> |---|---|---|---|---|
> | train loss | 2.361 | 2.355 | 2.365 | 2.359 |
> | train ppl | 10.61 | 10.55 | 10.65 | 10.59  |
> | final val loss | 2.371 | 2.363 | 2.377 | 2.367 |
> | final val ppl | 10.71 | 10.63 | 10.78 | 10.67 |
> | runtime | 15h 53m  | 15h 56m  | 17h 6m | 18h 17m |
> | memory usage (per GPU) | 149.9 GB | 149.9 GB | 173.87 GB | 175.52 GB |
> |
>
> **Weakness (2)**
>
> We would like to thank the reviewer for pointing this out. In our experiments we tried both $L_1$ and $L_2$ norms and observed they obtain similar results up to numerical precision errors. We considered that the $L_1$ norm is more intuitive to compute the similarities in matrix $S$ per row or per column by simply summing up the absolute values (e.g. the scalar products).
>
> The rank sensitivity can be seen in Table 1 in our paper for Dion, where the running time does not change with the rank, in contrast to the Dion optimizer. The same trend follows in the results for the Llama-7B model above.
>
> Regarding the DCT variant, we used DCT-III throughout all experiments. As our Dynamic Column Selection approach works with any orthogonal matrix, we can also use DCT-II (the transpose of DCT-III). However, it is important to mention that in the context of using Makhoul’s algorithm, we would need to use DCT-II matrix for the right projection (where we process rows of G) and DCT-III for the left projection (where we process columns of G) in order for the FFT-implementation to be equivalent with the matmul version.
>
> We will update our manuscript to include these clarifications.
>
> **Weakness (3)**
>
> In our work we provided results only in DDP settings. We are currently working on an implementation for FSDP-2 that is built on top of the Muon implementation from **[1]**. It splits the gradient matrices into buckets with $n$ matrices, where $n$ is the total number of GPUs (world size), uses all-to-all communication to materialize the full-matrices on each GPU and then applies our Dynamic Column Selection algorithm to compute low-rank compression followed by orthogonalization via Newton-Schulz. Once we obtain the orthogonal, low-rank momentum, all-to-all is called again on the lower dimensional tensor to shard the parameters. We will release our implementation with the next revision of our paper as we have some more work to do around sharding the DCT matrix. We need to do a bit more work towards figuring out what is the most efficient strategy to shard one layer and the DCT matrix based on the projection type (left or right) such that computing the column similarities $S=GQ$ does not involve additional communication that is unnecessary.
>
> **References:**
>
> **[1]** Distributed Muon using all-to-all via https://github.com/microsoft/dion/blob/main/dion/muon.py
>
> **Weakness (4)**
>
> We thank the reviewer for mentioning these two prior works. Due to limited time and reduced computational budget, we provide some comparison of DCT-AdamW and SubTrack++ on a model with 350M parameters, rank 256 (25% of model’s embedding size) and preconditioner update frequency 200. We run three experiments with different random seeds for each optimizer and report the average metrics.
>
> | Metric                               | DCT-AdamW | SubTrack++ |
> |-----------------------------------|--------------------|-----------------|
> | train loss                           | 2.738             | 2.779           |
> | train ppl                             | 15.48             | 16.13           |
> | final val loss                      | 2.747             | 2.789           |
> | final val ppl                        | 15.60             | 16.266         |
> | runtime                              | 6h 57m          | 7h 15m        |
> | memory usage (per GPU) | 42.99 GB       | 46.95 GB    |
> |
>
> We observe that our DCT-AdamW achieves significantly lower values for train and validation metrics with slightly faster runtime and memory usage, which proves that our Dynamic Column Selection approach is effective in tracking the low-rank subspace of the gradient.

---

> ### Author Response · Authors · 2025-11-26
>
> Dear reviewer,
>
> As we are getting closer to the end of the author-reviewer discussion period, we are kindly asking you to have a look at our answers to your questions and please let us know whether they clarified your concerns.
>
> Respectfully, The authors

---

### Official Review · Reviewer_akDN · 2025-11-04

**Soundness:** 3
**Presentation:** 3
**Contribution:** 2
**Rating:** 6
**Confidence:** 3

**Summary:**

The paper proposes replacing per step SVD or QR low rank projections in adaptive optimizers with a fixed orthogonal basis using DCT and dynamic column selection. At each step the method scores basis columns by gradient to basis correlation and selects the top r to form projections, which avoids repeated factorizations and heavy state storage. Two instances, Trion and DCT AdamW, show lower memory use, faster training, and comparable or better perplexity on mid size LLM pretraining. The theory motivates norm based selection and provides simple error bounds. The implementation also explains how to integrate with distributed training to reduce communication.

**Strengths:**

The paper uses a precomputed DCT with on the fly selection to replace repeated SVD or QR, and only column indices are stored. Column norm ranking aligns with minimizing reconstruction error, and there is clear intuition for why DCT approximates dominant directions.

The method has consistent memory reduction and wall clock speedups while matching or improving perplexity across several model sizes.

The notes on DDP or FSDP usage and local reconstruction make adoption straightforward.

**Weaknesses:**

Evidence for scaling to very large hidden sizes remains limited. The paper primarily reports results on mid size models where the benefit of fast transforms over plain matrix multiplication is muted. Without end to end wall clock measurements at dimensions around 8k to 16k, and without a breakdown of time in the similarity computation, basis selection, and reconstruction kernels, it is hard to assess whether the claimed speedups persist when layers become wide and deep. A thorough profiling study across model width, batch size, and rank would make the efficiency claims more convincing.

It is unclear whether all systems level optimizations such as ZeRO style redundancy removal, identical communication and precision settings, and error feedback or quantization choices are enabled symmetrically for both the proposed method and the baselines. Differences in these controls can easily dominate the observed speed or memory gains. The paper should report results under strictly matched configurations and, if desired, separately include best tuned variants for each method.

The paper mixes square and rectangular gradient matrices without clearly specifying when to apply left versus right projections, and how this choice is made across different layer types such as attention projections and output layers. The exact dimensional assumptions of the DCT matrices and the consistency of symbols drift across sections, which complicates reproduction and theoretical interpretation. A precise, layer wise rule set and a short ablation on these choices would improve clarity.

The set of comparative baselines is too narrow to establish superiority. Strong structured projections such as Hadamard or FWHT, CountSketch style mappings, and recent online or streaming low rank methods are not evaluated side by side under matched rank and refresh frequency. Because many of these alternatives also offer O(n log n) or near linear time with tiny memory footprints, omitting them leaves open whether DCT based selection is uniquely effective. Head to head comparisons on identical tasks and budgets are needed to justify the design choice.

**Questions:**

Please see weakness.

---

> ### Author Response · Authors · 2025-11-21
> **[1] Authors' answers for Weaknesses 1 & 2**
>
> We would like to thank the reviewer for their feedback. We address your concerns below.
>
> **Weakness (1)**
>
> We provide a detailed benchmarking on running time of optimizer state, please read our answers for **Weakness 3 of Reviewer #1 ZMSS** and **Weakness #1 of Reviewer #4 F7Ct**.
>
> Regarding larger scale experiments, we replicate the setup in Table 1 to train a Llama-7B model with two exceptions:
>
> 1) We use a node with 8x Nvidia B200 GPUs 180GB instead of 8x Nvidia H100 80 GB
>
> 2) We use only 10B tokens (equivalent to 1.5x model size) instead of 20x model size
>
> We are aware that this setup is suboptimal and we hope the reviewers are aware of the costs of training such a model using academic resources.
>
> By analyzing the results provided in the table below, our claim from the paper also holds at this large model scale: Trion exhibits the same running time and memory usage for different ranks. In contrast, the running time and memory usage of Dion depend on the rank because of the QR-decomposition procedure, which we successfully replaced in Trion with our cheaper Dynamic Columns Selection approach.
>
> | Metric | Trion $r=512$ | Trion $r=1024$ | Dion $r=512$ | Dion $r=1024$ |
> |---|---|---|---|---|
> | train loss             | 2.361    | 2.355    | 2.365     | 2.359     |
> | train ppl              | 10.61    | 10.55    | 10.65     | 10.59     |
> | final val loss         | 2.371    | 2.363    | 2.377     | 2.367     |
> | final val ppl          | 10.71    | 10.63    | 10.78     | 10.67     |
> | runtime                | 15h 53m  | 15h 56m  | 17h 6m    | 18h 17m   |
> | memory usage (per GPU) | 149.9 GB | 149.9 GB | 173.87 GB | 175.52 GB |
> |
>
>
> **Weakness (2)**
>
> Our optimizers and baselines inherit some characteristics of the ZeRO technique for DDP settings. Concretely, we do not replicate the optimizer states on all GPUs. Instead, we compute the model update for one layer on a single GPU and communicate the result to the other GPUs. This way, there are no redundant computations in our optimizer. This reduces memory as we do store optimizer states for one layer only on one GPU.
>
> We compare against the official DDP implementation of the Dion optimizer, where each device will orthogonalize one full matrix, identically to our Trion implementation described above.
>
> We make sure all settings are the same and the differences are only in the optimization algorithm itself.
>
> We would like to thank the reviewer for this observation as we believe it is crucially important for the correctness of the results and will clearly explain these details in the next revision of our paper.

---

> ### Author Response · Authors · 2025-11-21
> **[2] Authors' answers for Weaknesses 3 & 4**
>
> **Weakness (3)**
>
> We follow the standard projection approach introduced in GaLore, where for a gradient $G \in \mathbb{R}^{R \times C}$ ($R$ rows and $C$ columns) we choose to collapse the smallest dimension to rank $r$.
>
> For $R \geq C$ we apply a **right projection** and we choose the most appropriate $r$ columns from the DCT matrix $Q$ of shape $(C, C)$, resulting a projection matrix $P \in \mathbb{R}^{C \times r}$ and a low-rank gradient $g = GP \in \mathbb{R}^{R \times r}$.
>
> For $R < C$ , we apply a **left projection** with DCT matrix of shape $(R, R)$, resulting a projection matrix $P \in \mathbb{R}^{r \times R}$ and a low-rank gradient $g = PG \in \mathbb{R}^{r \times C}$.
>
> This set of rules applies to each layer by choosing the left or right projection based on the layer dimensions and this is consistent in our paper.
>
> We thank the reviewer for pointing this out and we will clarify this in the next revision of our paper to avoid any confusion regarding how we apply low-rank projections.
>
> Regarding differences in the notations across sections, we are going to review our manuscript one more time to search for some inconsistencies. We would appreciate it if you had time to point them out during the rebuttal.
>
> **Weakness (4)**
>
> The core of our work is the Dynamic Column Selection algorithm that can be used with almost any fixed orthogonal matrix. In particular, we choose DCT because we can get a speed-up in certain conditions using Makhoul’s algorithm. We focused on replacing the SVD and QR-decomposition with our dynamic approach and compared against the underlying methods that we aim to improve on, for example Dion, LDAdamW, Frugal and Fira variants. We believe that replacing the core part of target optimizers with our approach is sufficient to prove the effectiveness of our method. We would like to point out that our computational resources are limited and we could not test against all possible methods. Instead, we chose the most popular.
>
> We would like to thank the reviewer for mentioning CountSketch and Streaming Low-Rank methods, which we were not aware of during the preparation of our paper. While we found a reference for CountSketch (see **[1]** and **[2]** below), we could not find a reference for the Streaming Low-Rank methods and we would kindly ask the reviewer to provide some prior work. We will also include both these missing prior works in our related work section for fair acknowledgement.
>
> Regarding the Hadamard/FWHT matrix, despite being theoretically orthogonal, in practice it might not be the case for certain dimensions. In contrast, the DCT matrix exists for any size and this is one of the reasons why we chose it, in addition to the potential speedup via Makhoul's N-point algorithm based on FFT.
>
> **References:**
>
> **[1]** Spring et al., **Compressing gradient optimizers via count-sketches**, ICML 2019, https://proceedings.mlr.press/v97/spring19a/spring19a.pdf
>
> **[2]** Song et al., **Sketching for First Order Method: Efficient Algorithm for Low-Bandwidth Channel and Vulnerability**, ICML 2023, https://arxiv.org/pdf/2210.08371

---

> ### Author Response · Authors · 2025-11-26
>
> Dear reviewer,
>
> As we are getting closer to the end of the author-reviewer discussion period, we are kindly asking you to have a look at our answers to your questions and please let us know whether they clarified your concerns.
>
> Respectfully, The authors

---

### Official Review · Reviewer_ZMSS · 2025-11-05

**Soundness:** 3
**Presentation:** 3
**Contribution:** 2
**Rating:** 4
**Confidence:** 4

**Summary:**

Low-rank optimization can speed LLM training and cut optimizer memory, but per-layer SVD/QR gradient projections are costly and require storing projection matrices. We propose a simple DCT-based alternative: multiply each layer’s gradient by a fixed orthogonal DCT basis, then rank-select the most aligned columns to form the projection. The DCT is efficiently computed (via FFT-based routines), precomputed once, and reused—so projections need only a matmul plus lightweight sorting. Across pre-training and fine-tuning, this yields rank-independent runtime, matches SVD/QR accuracy, and achieves faster training with lower memory use.

**Strengths:**

- Provided a two-stage DCT-based projection method to avoide the computational cost of SVD
- Developed the DCT variant Trion and DCT-AdamW
- Demenstrated the contractivenss of the proposed compressor
- Conduct extensive experiments

**Weaknesses:**

1. **Effectiveness in tracking gradients.** Constructing SVD-free, contractive projection matrices is straightforward (e.g., random projections). The real challenge is to remain SVD-free and contractive *while* faithfully capturing the gradient’s low-rank structure. The proposed two-stage method fixes a DCT basis $D_C$ for the entire training and, at each iteration, selects only a few of its columns. In effect, it tracks gradients using subsets of a preset basis—an approach not obviously aligned with evolving, layer-specific low-rank subspaces. The paper does not explain why this selection should match the true gradient subspaces or under what conditions it would; clearer insight or evidence (e.g., principal-angle analyses or SVD/QR approximation errors over training) is needed.

2. **Necessity of compressing optimizer states.** ZeRO-style sharding already reduces optimizer-state memory by ~1/N per data-parallel replica without degrading quality. In Figure 2, DCTAdamW underperforms AdamW, suggesting that extra low-rank compression of states may be unnecessary—or even harmful—unless it delivers clear end-to-end gains. I personally think compressing optimizer states is unnecessary due to ZeRO optimizer.

3.  **Necessity of saving computations in Newton–Schulz.** In Trion/Muon-style preconditioning, forward–backward passes dominate step time; Newton–Schulz iterations are typically a small fraction—especially with long context windows. The paper should provide profiler traces showing that low-rank $b_t$ meaningfully reduces *end-to-end* step time or time-to-target. Please also compare with vanilla Muon (no low-rank) to assess any convergence slowdown from low-rank preconditioning, and include both complexity estimates (per-token FLOPs) and wall-clock measurements to substantiate the claimed benefit.

**Questions:**

1. **Error-feedback memory overhead**. Trion’s error-feedback buffer appears to store a full-size residual per parameter. Does this negate the memory savings from low-rank gradient projection?

2. **Low-rank during forward–backward**. Beyond optimizer/state compression, can the low-rank structure be exploited to reduce the dominant forward–backward costs (FLOPs and memory)—e.g., via factored weight updates or structured bases that lower gradient-computation cost?

3. I noticed that the work [R1] uses a similar idea to save SVD computations. Could the authors highlight the difference from [R1]

[R1] Wavelet Meets Adam: Compressing Gradients for Memory-Efficient Training

---

> ### Author Response · Authors · 2025-11-21
> **[1] Authors' answers to Weakness 1 & Weakness 2**
>
> We would like to thank the reviewer for their feedback. We address your concerns below.
>
> **Weakness (1)**
>
> The effectiveness of our Dynamic Column Selection approach lies in adaptively choosing the $r$ columns from a fixed basis of $n$ possible directions to minimize the projection error.
>
> We choose different columns from the fixed basis at each step and as the gradient changes over time, we obtain different projection matrices. We efficiently choose the subspace that yields the minimal projection error, as the gradient evolves over time.
>
> This changes the alignments with the columns of our orthogonal matrix of choice, resulting in the optimal subspace selection at each step with respect to our fixed orthogonal matrix.
>
> Indeed, using this approach we obviously incur larger projection error than SVD (which is the optimal projection, but way more expensive).
>
> In the submitted manuscript in Figure 1 (page 6, lines 290-295) we show the projection error for Trion and Dion, which we use as a clearer evidence of the effectiveness of our method compared to QR-decomposition. Note that Subspace Iteration (that calls QR-decomposition) used in Dion is indeed cheaper than SVD, but still more expensive than our approach and still doesn’t match the true gradient subspaces, since it is also an approximation.
>
> **Weakness (2)**
>
> We agree with the reviewer that ZeRO-style sharding already reduces the optimizer state memory per device, but in reality there is no compression performed. In our work we take into consideration the 2D structure of Linear layers in the Transformer blocks and therefore perform low-rank compression of the corresponding 2D matrices. We focus on reducing the overhead of SVD/QR-based techniques applied to the particular area of low-rank compression of optimizer states.
>
> Our optimizers inherit some characteristics of the ZeRO technique for DDP settings. Concretely, we do not replicate the optimizer states on all GPUs. Instead, we compute the model update for one layer on a single GPU and communicate the result to the other GPUs. This way, there are no redundant computations in our optimizer. This reduces memory as we do store optimizer states for one layer only on one GPU.
>
> Regarding Figure 2, AdamW is added just for the reference since LDAdamW and DCT-AdamW are a low-rank version of Adam. The comparison should mainly be with LDAdamW.

---

> ### Author Response · Authors · 2025-11-21
> **[2] Authors' answers to Weakness 3**
>
> **Weakness (3)**
>
> We thank the reviewer for this question. We created a benchmark to be able to isolate the running time per step of our Trion optimizer in comparison to Dion and Muon. We explain the concrete steps below. First, we provide further results for Muon and then provide the benchmarking for the optimizer step.
>
> **Comparison against Muon in the same settings as Table 1:**
>
> We replicate the setup presented in Table 1 and train Llama models with 350M, 800M and 1.3B parameters on 8xNvidia H100 GPUs using Muon (full rank) with exactly the same hyper-parameters and seeds.
>
> Full-rank Muon obtains validation loss/perplexity close to our Trion variant with rank 512. Concretely, the full-rank Muon is better than Trion with respect to the validation perplexity metric by 0.16 for 350M, 0.13 for 800M and 0.12 for the 1.3B model. The perplexity gap has a seemingly decreasing trend across all tested models. The running time of Muon is similar to our Trion implementation, the time differences being caused by the cluster usage.
>
> Note that our work does not focus on communicating low-rank gradients in between the GPUs, which has the potential of reducing the overall runtime in distributed settings. We are currently working on implementing an efficient low-rank communication procedure, as the results show that our Dynamic Column Selection is an effective approach.
>
> | Metric | Llama-350M | Llama-800M | Llama-1.3B |
> |---|---|---|---|
> | train loss | 2.697 | 2.517 | 2.513 |
> | train ppl  | 14.85 | 12.41 | 12.35 |
> |
> | val loss | 2.707 | 2.489 | 2.409 |
> | val ppl  | 14.99 | 12.05 | 11.13 |
> |
> | runtime  | 1h 52m   | 8h 25m   | 19h 17m  |
> | memory usage (per GPU)  | 42.42 GB | 67.45 GB | 63.64 GB |
> |
>
> **Benchmarking the running time of optimizer step**:
>
> First, we create a custom model with only one transformer block with embedding dimensionality 4096 and number of heads 32, resulting in a model with 464M parameters in total (embeddings, transformer block, lm-head). We then train this model on a single GPU with sequence length 256 and batch size 1 to minimize the time spent in forward, backward and state communication, as well as the memory usage.
>
> Second, we used the model with 800M parameters from Table 1 in our paper and trained it on one GPU with sequence length 512 and batch size 1 to replicate the previous setup.
>
> For both scenarios, we measure and report the running time of the optimizer step. This way, we want to show the overhead of low-rank variants with respect to standard, full-rank Muon.
>
> We test full-rank Muon and low-rank optimizers Dion and Trion (with matmul and Makhoul’s algorithm) with rank 512 and 1024 to show the overhead of each setting. We perform experiments on one GPU of type Nvidia H100 and Nvidia RTX 3090 because H100 has fast tensor cores that make matmul faster, while RTX 3090 has slower tensor cores that benefit Makhoul’s algorithm, as shown in the benchmark we posted in our answer for *Weakness #1* for the *Reviewer #4 F7Ct*.
>
> Below we present the results for this synthetic benchmark that emphasizes the overhead of the optimizer step for Trion and Dion in comparison to full-rank Muon.
>
> Comments:
>
> - in **Rebuttal Table 1** (next comment) we show the benchmarking results for Llama-800M on H100. We observe that matmul is faster than Makhoul’s algorithm for both ranks, which is expected because H100 has fast tensor cores. Trion achieves lower running time than the full-rank Muon, while Dion is very expensive because of the QR-decomposition and our Dynamic Column Selection successfully replaces it.
>
> - in **Rebuttal Table 2** (next comment) we show the benchmarking results for Llama-800M on RTX-3090. We observe that matmul is slower than Makhoul’s algorithm for both ranks, which is explained by the slower tensor-cores for this older GPU. Trion achieves lower running time than full-rank Muon for both ranks.
>
> - in **Rebuttal Table 3** (next comment) we show the benchmarking results for our custom Llama-464M on H100. As in the previous case, matmul is faster on H100 than Makhoul’s algorithm. Since the layers are larger, we observe some increase in the overhead for low-rank methods. Still, Trion is much cheaper than Dion, whose running time increases significantly with the rank.
>
> - in **Rebuttal Table 4** (next comment) we benchmark our custom Llama-464M on RTX-3090. Similarly, Makhoul is slightly faster than matmul and the low-rank methods have an additional overhead compared to Muon. Still, Dion is the most expensive also in this case, which is justified by the calls to QR-decomposition.

---

> ### Author Response · Authors · 2025-11-21
> **[3] Authors' answers to Weakness 3 (continued)**
>
> **Rebuttal Table 1: [H100] Llama-800M (d=2048, layers=16, batch-size=1, seq_len=512)**
>
> | Method | Rank | OptimizerStep (ms/it) | TotalMemUsage (GB) | Time Overhead vs Muon | Memory Overhead vs Muon |
> |---|---|---|---|---|---|
> | Muon            | full | 113 | 19.07 | 1x | 1x |
> |
> | Trion+matmul    | 512 |  67  | 18.61 | 0.59x | 0.98x |
> | Trion+Makhoul   | 512 |  78  | 19.02 | 0.69x | 1x |
> | Dion            | 512 | 246  | 20.47 | 2.18x | 1.07x |
> |
> | Trion+matmul    | 1024 |  90 | 18.87 | 0.80x  | 0.99x |
> | Trion+Makhoul   | 1024 | 101 | 19.02 | 0.89x | 1x |
> | Dion            | 1024 | 570 | 20.65 | 5.04x | 1.08x |
> |
>
> **Rebuttal Table 2: [RTX-3090] Llama-800M (d=2048, layers=16, batch-size=1, seq_len=512)**
>
> | Method | Rank | OptimizerStep (ms/it) | TotalMemUsage (GB) | Time Overhead vs Muon | Memory Overhead vs Muon |
> | ---|---|---|---|---|---|
> | Muon           | full | 830 |  18.46 | 1x | 1x |
> |
> | Trion+matmul   | 512  |  355 | 19.06 | 0.43x | 1.03x |
> | Trion+Makhoul  | 512  |  315 | 18.66 | 0.38x | 1.01x |
> | Dion           | 512  |  485 | 19.59 | 0.58x | 1.06x |
> |
> | Trion+matmul   | 1024 |  551 | 19.06 | 0.66x | 1.03x |
> | Trion+Makhoul  | 1024 |  507 | 18.66 | 0.61x | 1.01x |
> | Dion           | 1024 | 1010 | 19.84 | 1.22x | 1.07x |
> |
>
> **Rebuttal Table 3: [H100] Llama-464M (d=4096, layers=1, batch-size=1, seq_len=256)**
>
> | Method | Rank  | OptimizerStep (ms/it) |  TotalMemUsage (GB) | Time Overhead vs Muon | Memory Overhead vs Muon |
> |---|---|---|---|---|---|
> | Muon           | full  |  3.19 | 13.01 | 1x  | 1x  |
> |
> | Trion+matmul   | 512   |  4.07 | 13.03 | 1.28x | 1x  |
> | Trion+Makhoul  | 512   |  4.41 | 13.48 | 1.38x | 1.04x |
> | Dion           | 512   | 11.95 | 15.36 | 3.75x | 1.18x |
> |
> | Trion+matmul   | 1024  |  4.02 | 13.03 | 1.26x | 1x  |
> | Trion+Makhoul  | 1024  |  4.45 | 13.48 | 1.39x | 1.04x |
> | Dion           | 1024  | 22.60 | 15.63 | 7.08x | 1.2x  |
> |
>
> **Rebuttal Table 4: [RTX-3090] Llama-464M (d=4096, layers=1, batch-size=1, seq_len=256)**
>
> | Method | Rank  |  OptimizerStep (ms/it) |  TotalMemUsage (GB) | Overhead vs Muon | Memory Overhead vs Muon |
> |---|---|---|---|---|---|
> | Muon           | full  |  6.70 | 12.98 | 1x  | 1x |
> |
> | Trion+matmul   | 512   |  8.01 | 12.37 | 1.2x  | 0.95x |
> | Trion+Makhoul  | 512   |  7.89 | 12.95 | 1.18x | 1x    |
> | Dion           | 512   | 13.26 | 14.58 | 1.98x | 1.12x |
> |
> | Trion+matmul   | 1024  |  7.01 | 12.37 | 1.05x | 0.95x |
> | Trion+Makhoul  | 1024  |  5.78 | 12.95 | 0.86x | 1x    |
> | Dion           | 1024  | 57.29 | 14.82 | 8.55x | 1.14x |
> |

---

> ### Author Response · Authors · 2025-11-21
> **[4] Authors' answers to Questions 1, 2, 3**
>
> **Question (1)**
>
> Our Trion optimizer follows the schematic of Dion and the only difference lies in how the low-rank projection is computed. Note the error feedback $\Delta_t$ is not explicitly stored in between consecutive steps and it only exists in the pseudocode for clarity purposes. In the original Dion implementation, the momentum term is updated as $M_t \gets B_t - (1 - \mu) P_t R_t^\top$ (see step 7 in Algorithm 1 in **[1]** and implementation **[2]**), so there is no error feedback that can negate the memory savings from low-rank gradient projection. The matrices $P_t$ and $R_t$ represent the optimizer state and they indeed increase the memory usage for Dion.
>
> We implement the error feedback in the same way in our Trion variant: please check lines 9-10 in Algorithm 3 at lines 182-183 our manuscript: we define $\Delta_t$ in line 9, but in line 10 we update the momentum as $M_t \gets B_t − (1 − \mu)b_t Q^\top$. To conclude, the error feedback does not negate the memory savings in practice.
>
> **References:**
>
> **[1]** Dion algorithm: https://arxiv.org/pdf/2504.05295v2#page=3
>
> **[2]** Dion implementation: https://github.com/microsoft/dion/blob/main/dion/dion.py#L847
>
> **Question (2)**
>
> We thank the reviewer for the question, this is actually a relevant one in the context of low-rank optimization.
>
> There is already some prior work in the literature that focuses on reducing the memory usage of activations by projecting the weights to a lower-dimensional space. This technique is called Randomized Subspace Optimization (RSO) **[1]** and it optimizes a low-rank weight matrix $B_t$ instead of full-sized weights $W_t$. At each step, they use a projection matrix $P_t$ to switch from the lower-dimensional space to the higher-dimensional space.
>
> One additional direction is to use low-rank compression for gradients in Distributed Data Parallel (DDP) settings and the most influential work in this direction is PowerSGD **[2]**. This technique compresses and decompresses the gradients before and after all-reduce alleviate the communication overhead.
>
> In contrast, our work focuses on reducing the overhead of SVD/QR-based low-rank projections for compressing the optimizer states.
>
> **References:**
>
> **[1]** Chen et al., **A Memory Efficient Randomized Subspace Optimization Method for Training Large Language Models**, ICML 2025, link https://arxiv.org/pdf/2502.07222
>
> **[2]** Vogels et al., **PowerSGD: Practical Low-Rank Gradient Compression for Distributed Optimization**, NeurIPS 2019, link https://arxiv.org/pdf/1905.13727
>
> **Question (3)**
>
> Conceptually, Wavelet Adam [R1] has some similarities to our approach. In particular, they also use a fixed orthogonal matrix (Haar wavelet transform) to project the gradients before passing to a state-full optimizer like Adam.
>
> However, the way this fixed orthogonal matrix is employed and how the projection error is handled are different. After orthogonal transformation, they always take the first half of the signal as the projection “Approximation” and classify the rest as “Detail”. One can easily come up with a (worst-case) gradient signal for which the approximation is 0, while the “Detail” part is the entire gradient.
>
> In our approach, after DCT transformation we dynamically select the most representative part of the gradient signal. Dynamic Column Selection makes sure that projected gradient contains information at least proportional to the number of components.
>
> In their approach, the projection error or “Detail” is incorporated into the update using Adam’s preconditioning built on top of the “Approximation” part of the gradient. In contrast, we handle the projection error following the well-established Error Feedback framework.
>
> Another difference on the practical side is that they use norm-growth limiter to handle loss spikes during the training.

---

> ### Author Response · Authors · 2025-11-26
>
> Dear reviewer,
>
> As we are getting closer to the end of the author-reviewer discussion period, we are kindly asking you to have a look at our answers to your questions and please let us know whether they clarified your concerns.
>
> Respectfully, The authors

---

### Meta-Review · Area_Chair_9axT · 2026-01-05

**Summary:**

The paper proposes replacing per-step SVD or QR–based low-rank projections in adaptive optimizers with a fixed orthogonal basis constructed via the DCT, combined with dynamic column selection, to reduce memory usage and improve runtime efficiency for these optimizers.

Prior to the rebuttal, reviews were mixed. The primary concern was whether Trion provides clear advantages over Dion or Muon at larger model scales, and where these benefits come from. In the rebuttal, the authors present additional experiments to strengthen their claims. In particular, results on LLaMA-3 8B are used to demonstrate scalability, and optimizer step-time measurements show the practical benefits of Makhoul’s algorithm on certain older GPU architectures.

Some concerns remain. For smaller models, the additional results indicate that Muon outperforms Trion despite comparable memory costs and overall runtime. Moreover, the paper does not study alternative design choices, such as different orthogonal bases, particularly given that the benefits of Makhoul’s algorithm appear marginal on modern GPUs (e.g., H100). In addition, the original Dion paper reports clear per-step time savings over Muon for large-scale models such as LLaMA-3 405B, and includes benchmarks across a range of square matrix dimensions. No similar results are provided in Trion.

After carefully reading the paper, the reviews, and the rebuttal, the Area Chair recommends accepting the paper as a poster presentation, as the potential efficiency gains of the proposed method appear promising. The authors are encouraged to incorporate the additional results from the rebuttal into the camera-ready version.

**Reviewer Concerns:**

The major concerns from the reviewers are how Trion performs on larger-scale models. This concern should be resolved with the results of the LLaMA-3 8B model. The authors also addressed concerns regarding scaling to larger hidden sizes, the runtime of the optimizer step, and additional comparison baselines.

The remaining concerns include scaling to very large models, as shown in the Dion paper, the effectiveness of Makhoul’s algorithm on advanced GPUs, missing FSDP-2 implementation, and alternative design choices.

**Reviewer Scores:**

Reviewer ZMSS would improve his scores to 6 since most of the concerns have been resolved. Reviewer tJpX would maintain his score since the design choice and FSDP-2 implementations are missing. Other reviewers will maintain their positive ratings.

---

### Decision · Program_Chairs · 2026-01-26

Accept (Poster)